# Unifying Generation and Prediction on Graphs with Latent Graph Diffusion

**Cai Zhou**[1,2], **Xiyuan Wang**[3,4], **Muhan Zhang**[3*]

[1]Department of Automation, Tsinghua University
[2]Department of Electrical Engineering and Computer Science, Massachusetts Institute of Technology
[3]Institute for Artificial Intelligence, Peking University
[4]School of Intelligence Science and Technology, Peking University
`caiz428@mit.edu`, `{wangxiyuan,muhan}@pku.edu.cn`

## Abstract

In this paper, we propose the first framework that enables solving graph learning tasks of all levels (node, edge and graph) and all types (generation, regression and classification) using one formulation. We first formulate prediction tasks including regression and classification into a generic (conditional) generation framework, which enables diffusion models to perform deterministic tasks with provable guarantees. We then propose Latent Graph Diffusion (LGD), a generative model that can generate node, edge, and graph-level features of all categories simultaneously. We achieve this goal by embedding the graph structures and features into a latent space leveraging a powerful encoder and decoder, then training a diffusion model in the latent space. LGD is also capable of conditional generation through a specifically designed cross-attention mechanism. Leveraging LGD and the "all tasks as generation" formulation, our framework is capable of solving graph tasks of various levels and types. We verify the effectiveness of our framework with extensive experiments, where our models achieve state-of-the-art or highly competitive results across a wide range of generation and regression tasks.

## 1 Introduction

Graph generation has become of great importance in recent years, with important applications in many fields, including drug design [Li et al., 2018] and social network analysis [Grover et al., 2019]. However, compared with the huge success of generative models in natural language processing [Touvron et al., 2023] and computer vision [Rombach et al., 2021], graph generation is faced with many difficulties due to the underlying non-Euclidean topological structures. Existing models fail to generate structures and features simultaneously [Niu et al., 2020, Jo et al., 2022], or model arbitrary attributes [Vignac et al., 2022].

Moreover, while general purpose foundation models are built in NLP [Bommasani et al., 2021] which can solve all types of tasks through the generative framework, there are no such general purpose models for graph data, since current graph generative models cannot address regression or classification tasks. People also have to train separate models for different levels (node, edge, and graph) of graph learning tasks. Therefore, it is beneficial and necessary to build a framework that can solve (a) tasks of all types, including generation, classification and regression; (b) tasks of all levels, including node, edge, and graph-level.

---

*Corresponding Author.

38th Conference on Neural Information Processing Systems (NeurIPS 2024).

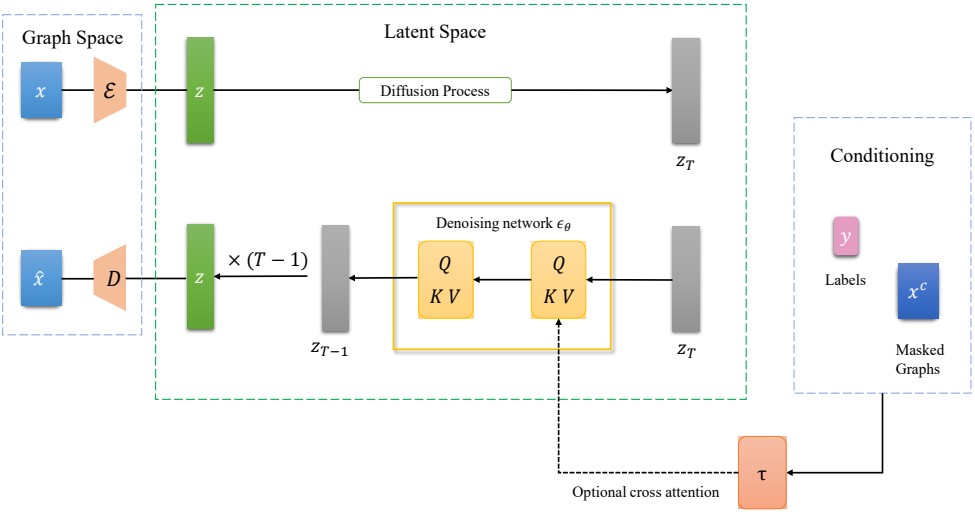

Figure 1: Illustration of the Latent Graph Diffusion framework, which is capable of performing both generation and prediction.

**Our work.** To overcome the difficulties, we propose Latent Graph Diffusion, **the first graph generative framework that is capable of solving tasks of various types and levels**. Our main contributions are summarized as follows.

- We conceptually formulate regression and classification tasks as conditional generation, and theoretically show that latent diffusion models can address them with provable guarantees.
- We present Latent Graph Diffusion (LGD), which applies a score-based diffusion generative model in the latent space encoded through a powerful pretrained graph encoder, thus overcoming the difficulties brought by discrete graph structures and various feature categories. Moreover, leveraging a specially designed graph transformer, LGD is able to generate node, edge, and graph features simultaneously in one shot with all feature types. We also design a special cross-attention mechanism to enable controllable generation.
- Combining the unified task formulation as generation and the powerful latent diffusion models, LGD is able to perform both generation and prediction tasks through generative modeling.
- Experimentally, LGD achieves state-of-the-art or highly competitive results across various tasks including molecule (un)conditional generation as well as regression and classification on graphs. Our code is available at https://github.com/zhouc20/LatentGraphDiffusion.

## 2   Related Work

**Diffusion models.** Recently, score-based generative models have demonstrated promising performance across a wide range of generation tasks. These models start by introducing gradually increasing noise to the data, and then learn the reverse procedure by estimating the score function, which represents the gradient of the log-density function with respect to the data. This process allows them to generate samples effectively. Two notable works in the realm of score-based generative models are score-matching with Langevin dynamics (SMLD) [Song and Ermon, 2019] and the denoising diffusion probabilistic model (DDPM) [Ho et al., 2020]. SMLD estimates the score function at multiple noise scales and generates samples using annealed Langevin dynamics to reduce the noise. In contrast, DDPM models the diffusion process as a parameterized Markov chain and learns to reverse the forward diffusion process of adding noise. Song et al. [2020b] encapsulates SMLD and DDPM within the framework of stochastic differential equations (SDE). Song and Ermon [2020] and Song et al. [2020a] further improved the scalability and sampling efficiency of score-based generative methods. More recently, Stable-Diffusion [Rombach et al., 2021] applied diffusion models to the latent space encoded by pretrained autoencoders, significantly improving computational efficiency and the quality of generated images. Our work has further applied latent space to graph tasks and also improved the generation quality.

**Diffusion models for graphs.** In the context of graph data, Niu et al. [2020] were the first to generate permutation-invariant graphs using score-based methods. They achieved this by introducing Gaussian perturbations to continuous adjacency matrices. Jo et al. [2022] took this step further by proposing a method to handle both node attributes and edges simultaneously using stochastic differential equations (SDEs). However, these diffusion models rely on continuous Gaussian noise and do not align well with discrete graph structures. To address this limitation, Haefeli et al. [2022] designed a diffusion model tailored to unattributed graphs and observed that discrete diffusion is beneficial for graph generation. Addressing this issue even more effectively, DiGress [Vignac et al., 2022], one of the most advanced graph generative models, employs a discrete diffusion process. This process progressively adds discrete noise to graphs by either adding or removing edges and altering node categories. Additionally, Limnios et al. [2023] proposes a method to scale DiGress by sampling a covering of subgraphs within a divide-and-conquer framework. To overcome the difficulties, instead of utilizing discrete noise, our work uses continuous noise in latent space, making our model structure well aligned with general diffusion models and achieves better performance. Our generative framework supports both DDPM and SMLD. Given their equivalence in SDE formulations [Song et al., 2020b], we focus on DDPM and its extensions in our main text for illustration. More details on SMLD-based methods and SDE formulations can be found in Appendix A.

## 3 Regression and Classification as Conditional Generation

In this section, we use the notation $\mathbf{x}$ to denote the known data, and $\mathbf{y}$ to denote the labels to be predicted in the task. We use this unified notation not only for simplicity; actually, the formulation in this subsection is **generic** and not limited to graph data. Our framework is the first to show that **diffusion models can provably solve regression and classification tasks**.

From a high-level perspective, unconditional generation aims to model $p(\mathbf{x})$, while conditional generation aims to model $p(\mathbf{x}|\mathbf{y})$. For classification and regression tasks, traditional deterministic models give point estimates to minimize classification or regression loss. However, from a generative perspective, predicting a point estimation is not the only solution - we can also model the complete conditional distribution $p(\mathbf{y}|\mathbf{x})$. Since modeling (conditional) distribution is exactly what generative models are capable of (and good at), it is possible to solve classification and regression tasks with generative models.

Actually, the classification and regression tasks can be viewed as **conditional generation tasks**. Intuitively, we only need to exchange the positions of data $\mathbf{x}$ and $\mathbf{y}$ in the conditional diffusion model; then we obtain a diffusion model that approximates $p(\mathbf{y}|\mathbf{x})$. In this case, we use the data $\mathbf{x}$ as the condition and want to generate $\mathbf{y}$ given the condition $\mathbf{x}$. We add Gaussian noise to $\mathbf{y}$ in the diffusion process, and we use the denoising network to reverse the process. In the parameterization where the denoising network $\epsilon_\theta$ directly outputs the target instead of the noise analogous to Equation (8), i.e. $\hat{\mathbf{y}} = \epsilon_\theta(\mathbf{y}_t, t, \mathbf{x})$, then the diffusion objective is

$$\mathcal{L}(\epsilon_\theta) := \mathbb{E}_{\mathbf{x},\mathbf{y},t}\left[||\epsilon_\theta(\mathbf{y}_t, t, \mathbf{x}) - \mathbf{y}||_2^2\right] \tag{1}$$

It is straightforward to extend the above method to latent diffusion models, where the only difference is that the diffusion model operates in the latent space of $\mathbf{x}, \mathbf{y}$. Since the formulation is generic, the method is also applicable to graph data; see Section 5.

Now we present our main theorem on the generalization bound of solving regression tasks with latent diffusion models, while more details on formal formulation and proof are given in Appendix B.

**Theorem 3.1.** *Suppose Assumption B.1, Assumption B.2, Assumption B.3 hold, and the step size $h := T/N$ satisfies $h \preceq 1/L$ where $L \geq 1$. Then the mean absolute error (MAE) of the conditional latent diffusion model in the regression task is bounded by*

$$\text{MAE} \leq \mathbb{E}_q[||w^\top \hat{\tau}(\boldsymbol{x}, y) - w^\top \tau(\boldsymbol{x}, y)||] + \epsilon_1 \tag{2}$$

$$\preceq \underbrace{\sqrt{\text{KL}(q_z||\gamma^d)} \exp(-T)}_{\text{convergence of forward process}} + \underbrace{(L\sqrt{dh} + L(m_1 + m_2)h)\sqrt{T}}_{\text{discretization error}} + \underbrace{\epsilon_{\text{score}}\sqrt{T}}_{\text{score estimation error}} + \underbrace{\epsilon_1}_{\text{encoder error}} \tag{3}$$

*where $q_z$ is the ground truth distribution of $\boldsymbol{z} := w^t\big(\tau(\boldsymbol{x}, y) - \mathcal{E}(\boldsymbol{x})\big)$.*

Converting deterministic tasks such as regression and classification into generation has several potential advantages. First, generative models are capable of modeling the entire distribution, thus being able to provide information confidence interval or uncertainty by multiple predictions; in comparison, traditional deterministic models can only give point estimates. Secondly, generative models are often observed to have larger capacity and generalize well on large datasets [Kadkhodaie et al., 2024]. Specifically for latent diffusion models, since the generative process is performed in the expressive latent space, the training of diffusion models could be much faster than training a deterministic model from scratch, and we obtain a powerful representation of the data that can be used in other downstream tasks. Moreover, now that the deterministic tasks could be unified with generation, thus can be simultaneously addressed by one unified model, which is the precondition of training a foundation model. We refer readers to Section 6 and Appendix C for detailed experimental observations.

## 4    Latent Graph Diffusion

As discussed in Section 1 and Section 2, existing graph diffusion models fail to generate both graph structures and graph features simultaneously. In this section, we introduce our novel **Latent Graph Diffusion (LGD)**, a powerful graph generation framework which is the first to enable **simultaneous structure and feature generation** in one shot. Moreover, LGD is capable of generating features of **all levels** (node-level, edge-level and graph-level) and **all types** (discrete or categorical, continuous or real-valued).

**Notations.**    Given $n \in \mathbb{N}$, let $[n]$ denote $\{1, 2, 3, ..., n\}$. For a graph $G$ with $n$ nodes, we denote the node set as $v_i, i \in [n]$, and represent the node feature of $v_i$ as $\boldsymbol{x}_i \in \mathbb{R}^{d_v}$. In the context of node pairs $(v_i, v_j)$, there can be either an edge $e_{ij}$ with its edge feature $\boldsymbol{e}_{ij} \in \mathbb{R}^{d_e}$ or no edge connecting $v_i$ and $v_j$. To jointly model the structures and features (of node, edge, graph level), we treat the case where no edge is observed between $v_i$ and $v_j$ as a special edge type. We denote this as $e_{ij}$ with an augmented feature $\boldsymbol{e}_{ij} = \boldsymbol{e}' \in \mathbb{R}^{d_e}$, where $\boldsymbol{e}'$ is distinct from the features of existing edges, resulting in a total of $n^2$ such augmented edges. We represent all node features as $\boldsymbol{X} \in \mathbb{R}^{n \times d_v}$ where $\boldsymbol{X}_i = \boldsymbol{x}_i$, and augmented edge features as $\boldsymbol{A} \in \mathbb{R}^{n \times n \times d_e}$ where $\boldsymbol{A}_{i,j} = \boldsymbol{e}_{ij}$ (adjacency matrix with edge features). Graph-level attributes, denoted $\boldsymbol{g} \in \mathbb{R}^{d_g}$, can be represented by a virtual node or obtained by pooling over node and edge attributes (see Section 4.2 for details). For brevity, we temporarily omit graph-level attributes in the following text. Note that all attributes can be of arbitrary types, including discrete (categorical) and continuous (real-valued).

### 4.1    Overview

Simultaneous generation of structures and features in the original graph space is challenging due to discrete graph structures and diverse graph features at various levels and types. Therefore, we apply our graph diffusion models in the latent space $\mathcal{H}$ rather than operating directly on the original graphs. Graphs are encoded in the latent space using a **powerful pretrained graph encoder** $\mathcal{E}_\phi$ parameterized by $\phi$. In this latent space, all $n$ nodes and $n^2$ "augmented edges" have continuous latent representations. The node and edge representations are denoted as $\boldsymbol{H} = (\boldsymbol{Z}, \boldsymbol{W})$, where $\boldsymbol{Z} \in \mathbb{R}^{n \times d}$ and $\boldsymbol{W} \in \mathbb{R}^{n \times n \times d}$, with $d$ representing the hidden dimension.

$$\boldsymbol{H} = (\boldsymbol{Z}, \boldsymbol{W}) = \mathcal{E}_\phi(\boldsymbol{X}, \boldsymbol{A}) \tag{4}$$

To recover the graph structures and features from the latent space $\mathcal{H}$, we employ a **light-weight task-specific decoder** $\mathcal{D}_\xi$, which is designed to be lightweight and is parameterized by $\xi$. This decoder produces the final predicted properties of the graph for specific tasks, including node, edge, and graph-level attributes. It is important to note that we consider the absence of an edge as a specific type of edge feature. Furthermore, we can derive graph-level attributes from the predicted node features $\hat{\boldsymbol{X}}$ and edge features $\hat{\boldsymbol{A}}$, as explained in detail in Section 4.2.

$$(\hat{\boldsymbol{X}}, \hat{\boldsymbol{A}}) = \mathcal{D}_\xi(\hat{\boldsymbol{H}}) \tag{5}$$

To ensure the quality of generation, the latent space $\mathcal{H}$ must meet two key requirements: (a) it should be powerful enough to effectively encode graph information, and (b) it should be conducive to learning a diffusion model [Rombach et al., 2021], thus should not be too scattered nor high-dimensional.

Consequently, both the architectures and training procedures of the encoder $\mathcal{E}_\phi$ and the decoder $\mathcal{D}_\xi$ are of utmost importance. We will dive into the architecture and training details in Section 4.2.

After pretraining the encoder and decoder, we get a powerful latent space that is suitable for training a diffusion model. We can now train the generative model $\epsilon_\theta$ parametrized by $\theta$ operating on the latent space. As explained, the generative model could have arbitrary backbone, including SDE-based and ODE-based methods. For illustration, we still consider a Denoising Diffusion Probabilistic Model (DDPM) [Ho et al., 2020], while other generative methods can be analyzed similarly. In the forward diffusion process, we gradually add Gaussian noise to the latent representation $\boldsymbol{H} = (\boldsymbol{Z}, \boldsymbol{W})$ until the latent codes converge to Gaussian. In the backward denoising process, we use a denoising model parameterized by $\epsilon_\theta$ parametrized by $\theta$ to reconstruct the latent representation of the input graph, obtaining $\hat{\boldsymbol{H}}$. The network $\theta$ can be a general GNN or graph transformer, depending on tasks and conditions (see Appendix C.3). In the most general case, we use a graph transformer enhanced by expanded edges; see Section 4.3 for architecture and training details.

In conclusion, we have successfully overcome the challenges posed by discrete graph structures and diverse attribute types by applying a diffusion model in the latent space. Our model is the first to be capable of simultaneously generating all node, edge, and graph-level attributes in a single step. The overview of our latent graph diffusion is shown in Figure 1.

## 4.2 Autoencoding

**Architecture.** In principle, the encoder can adopt various architectures, including Graph Neural Networks (GNNs) and graph transformers. However, to generate structures and features at all levels effectively, the encoder must be able to represent node, edge, and graph-level features simultaneously. As discussed in Section 4.1, to generate samples of high quality, the latent space $\mathcal{H}$ should be both powerful and suitable for training a diffusion model. To fulfill these criteria, we employ a specially designed augmented edge-enhanced graph transformer throughout this paper, as elaborated in Section 4.3. In cases where input graph structures and features are complete, we also allow the use of message-passing-based Graph Neural Networks (MPNNs) and positional encodings (PEs) during the pretraining stage. However, these techniques may not be applicable for the denoising network, as the structures of the noised graphs become corrupted and ill-defined.

Since the latent space is already powerful, the decoder can be relatively simple. The decoder is task-specific, and we pretrain one for each task along with the encoder. In generation tasks, the decoder is responsible for recovering the structure and features of the graph. In classification tasks or regression tasks, it serves as a classifier or regression layer, respectively. In all cases, we set the decoder for each task as a single linear layer.

**Training.** Following Rombach et al. [2021] and Xu et al. [2023], we jointly train the encoder and decoder during the pretraining stage. This training process can be carried out in an end-to-end manner or similar to training autoencoders. The training objective encompasses a reconstruction loss and a regularization penalty.

$$\mathcal{L} = \mathcal{L}_{recon} + \mathcal{L}_{reg} \tag{6}$$

The decoder $\mathcal{D}$ reconstructs the input graph from the latent representation, denoted $(\hat{\boldsymbol{X}}, \hat{\boldsymbol{A}}) = \mathcal{D}(\boldsymbol{H}) = \mathcal{D}(\mathcal{E}(\boldsymbol{X}, \boldsymbol{A}))$. The specific form of the reconstruction loss $\mathcal{L}_{recon}$ depends on the type of data, such as cross-entropy or Mean Squared Error (MSE) loss. To force the encoder to learn meaningful representations, we consider the following reconstruction tasks: (i) reconstruct node features $\boldsymbol{X}$ from node representations $\boldsymbol{Z}$; (ii) reconstruct edge features $\boldsymbol{A}$ from edge representations $\boldsymbol{W}$; (iii) reconstruct edge features $\boldsymbol{e}_{ij}$ from representations of node $i$ and $j$; (iv) reconstruct node features $\boldsymbol{x}_i$ from edge representations $\boldsymbol{e}_{i.}$; (v) reconstruct tuples $(\boldsymbol{x}_i, \boldsymbol{x}_j)$ from edge representations $\boldsymbol{e}_{ij}$. When applicable, we opt to reconstruct node-level and edge-level positional encodings (PEs) according to the latent features. We can also employ masking techniques to encourage the encoder to learn meaningful representations. It is worth noting that downstream classification or regression tasks can be seen as a special case of reconstructing the masked graph; for more details, refer to Section 5.

To prevent latent spaces from having arbitrarily high variance, we apply regularization. This regularization can take the form of a KL-penalty aimed at pushing the learned latent representations towards a standard normal distribution (referred to as *KL-reg*), similar to the Variational Autoencoder

(VAE) [Kingma and Welling, 2013]. Alternatively, we can employ a vector quantization technique (referred to as *VQ-reg*), as seen in [van den Oord et al., 2017, Yang et al., 2023a].

## 4.3 Diffusion model

**Architecture.** Note that during both the forward and backward diffusion processes, we either add noise to the latent representations of nodes and edges or attempt to denoise them. When we need to generate the graph structures, the edge indexes are unknown, making the corrupted latent representations do not have a clear correspondence with the originaledge information. Consequently, message-passing neural networks (MPNNs) and similar methods are not applicable, as their operations rely on clearly defined edge indexes.

To address the ambiguity of edges in the noised graph, we require operations that do not depend on edge indexes. Graph transformers prove to be suitable in this context, as they compute attention between every possible node pair and do not necessitate edge indexes. However, we can still incorporate representations of augmented edges into the graph transformers through special design. To facilitate the joint generation of node, edge, and graph-level attributes, we adopt the most general graph transformer with augmented edge enhancement. We design a novel self-attention mechanism that could update node, edge and graph representations as described in Equation (58), see Appendix C.1 for details.

**Training and sampling.** Training the latent diffusion model follows a procedure similar to the one outlined in Appendix A, with the primary distinction being that training is conducted in the latent space. During this stage, both the encoder and the decoder remain fixed. Denoting $\boldsymbol{H}_0 = \mathcal{E}(\boldsymbol{X}, \boldsymbol{A})$, in the forward process, we progressively introduce noise to the latent representation, resulting in $\boldsymbol{H}_t$.

$$\mathcal{L}_{LGD}(\epsilon_\theta) := \mathbb{E}_{\mathcal{E}(\boldsymbol{X},\boldsymbol{A}),\epsilon_t \sim \mathcal{N}(\mathbf{0},\mathbf{I})}\left[||\epsilon_\theta(\boldsymbol{H}_t,t) - \epsilon_t||_2^2\right] \tag{7}$$

where we implement the function $\epsilon_\theta^{(t)}$ as described in Equation (18) using a time-conditional transformer, as detailed in Equation (58). There is also an equivalent training objective in which the model $\epsilon_\theta$ directly predicts $\boldsymbol{H}_0$ based on $\boldsymbol{H}_t$ and $t$, rather than predicting $\epsilon_t$.

$$\mathcal{L}_{LGD}(\epsilon_\theta) := \mathbb{E}_{\mathcal{E}(\boldsymbol{X},\boldsymbol{A})}\left[||\epsilon_\theta(\boldsymbol{H}_t,t) - \boldsymbol{H}_0||_2^2\right] \tag{8}$$

In the inference stage, we directly sample a Gaussian noise in the latent space $\mathcal{H}$, and sample $\boldsymbol{H}_{t-1}$ from the generative processes iteratively as described in Appendix A. After we get the estimated denoised latent representation $\hat{\boldsymbol{H}}_0$, we use the decoder to recover the graph from the latent space, which finishes the generation process.

## 4.4 Conditional generation

Similarly to other generative models [Rombach et al., 2021], our LGD is also capable of controllable generation according to given conditions $\boldsymbol{y}$ by modeling conditional distributions $p(\mathbf{h}|\boldsymbol{y})$, where $\mathbf{h}$ is the random variable representing the entire latent graph representation for simplicity. This can be implemented with conditional denoising networks $\epsilon_\theta(\mathbf{h}, t, \boldsymbol{y})$, which take the conditions as additional inputs. Encoders and decoders can also be modified to take $\boldsymbol{y}$ as additional input to shift latent representations towards the data distribution aligned with the condition $\boldsymbol{y}$.

**General conditions.** Formally, we preprocess $\boldsymbol{y}$ from various data types and even modalities (e.g. class labels or molecule properties) using a domain specific encoder $\tau$, obtaining the latent embedding of the condition $\tau(\boldsymbol{y}) \in \mathbb{R}^{m \times d_\tau}$, where $m$ is the number of conditions and $d_\tau$ the hidden dimension. The condition could be embedded into the score network through various methods, including simple addition or concatenation operations [Xu et al., 2023] and cross-attention mechanisms [Rombach et al., 2021]. We provide details of the cross-attention mechanisms in Appendix C.1.

The conditional LGD is learned via

$$\mathcal{L}_{LGD} := \mathbb{E}_{\mathcal{E}(\boldsymbol{X},\boldsymbol{A}),\boldsymbol{y},\epsilon_t \sim \mathcal{N}(\mathbf{0},\mathbf{I}),t}\left[||\epsilon_\theta(\boldsymbol{H}_t,t,\tau(\boldsymbol{y}) - \epsilon_t||_2^2\right] \tag{9}$$

if the denoising network is parameterized to predict the noise; $\epsilon_\theta$ can also predict $\boldsymbol{H}_0$ conditioning on $\boldsymbol{H}_t, t, \tau(\boldsymbol{y})$ analogous to Equation (8).

**Masked graph.** Consider the case where we have a graph with part of its features known, and we want to predict the missing labels which can be either node, edge, or graph level. For example, in a molecule property prediction task where the molecule structures and atom/bond types are known, we want to predict its property. This can be modeled as a problem of predicting a masked graph-level feature (molecule property) given the node and edge features. In this case, the condition is the latent representation of a graph that is partially masked. We denote the features of a partially masked graph as $(\boldsymbol{X}^c, \boldsymbol{A}^c, \boldsymbol{g}^c)$, where the superscript $c$ implies that the partially masked graph is the "condition", $\boldsymbol{X}_i^c = \boldsymbol{x}_i^c, \boldsymbol{A}_{i,j}^c = \boldsymbol{e}_{ij}^c, \boldsymbol{g}^c$ are observed node, edge and graph-level features respectively. The missing labels we aim to predict are denoted as $\boldsymbol{y}$, then the complete graph $(\boldsymbol{X}, \boldsymbol{A}, \boldsymbol{g})$ can be recovered by $(\boldsymbol{X}^c, \boldsymbol{A}^c, \boldsymbol{g}^c)$ and $\boldsymbol{y}$. Intuitively, this is similar to image inpainting, where we hope to reconstruct the full image based on a partially masked one. Therefore, we can model the common classification and regression tasks as a conditional generation, where we want to predict the label $\boldsymbol{y}$ (or equivalently, the full graph feature $(\boldsymbol{X}, \boldsymbol{A}, \boldsymbol{g})$) given the condition $(\boldsymbol{X}^c, \boldsymbol{A}^c, \boldsymbol{g}^c)$, see Section 5 for formal formulations. In this case, as the condition is a graph as well (though it might be partially masked), we propose a novel cross-attention for graphs as shown in Equation (60), see Appendix C.1 for more details.

## 5 Unified Task Formulation as Generation

Based on graph generative model on the latent space, we can address generation tasks of all levels (node, edge, and graph) using one LGD model. In this section, we detail how tasks of different types can be formulated as generation, thus becoming tractable using one generative model.

Now we summarize our Latent Graph Diffusion framework, which is able to (1) solve tasks of all types, including generation, regression, and classification through the framework of graph generation; and (2) solve tasks of all levels, including node, edge and graph level. See Figure 1 for illustration.

First, since LGD is an internally generative model, it is able to perform unconditional and conditional generation as discussed in Section 4. For nongenerative tasks including regression and classification, we formulate them into conditional generation in Section 3, thus can also be solved by our LGD model. In particular for graph data, we want to predict the labels $\boldsymbol{y}$ which can be node, edge, or graph level and of arbitrary types, given the condition of a partially observed graph $\boldsymbol{X}^c, \boldsymbol{A}^c, \boldsymbol{g}^c$. To better align with the second target of full graph generation, we model this problem as predicting $p(\boldsymbol{X}, \boldsymbol{A}, \boldsymbol{g} | \boldsymbol{X}^c, \boldsymbol{A}^c, \boldsymbol{g}^c)$, where $(\boldsymbol{X}, \boldsymbol{A}, \boldsymbol{g}) = (\boldsymbol{X}^c, \boldsymbol{A}^c, \boldsymbol{g}^c, \boldsymbol{y})$ is the full graph feature that combines the observed conditions and the labels to be predicted. In other words, the condition can be viewed as a partially masked graph whose masked part is the labels $\boldsymbol{y}$, and our task becomes graph inpainting, i.e. generate full graph features conditioning on partially masked features.

Now we have formulated all tasks as generative modeling. The feasibility of the second goal to predict all-level features is naturally guaranteed by the ability of our augmented-edge enhanced transformer architecture described in Section 4. We leave the discussion of the possibility of solving tasks from different domains to Appendix D.2.

## 6 Experiments

In this section, we use extensive experiments that cover tasks of different types (regression and classification) and levels (node, edge and graph) to verify the effectiveness of LGD. We first conduct experiments on traditional generation tasks to verify LGD's generation quality. We consider both unconditional generation and conditional generation tasks. We then show that utilizing our unified formulation, LGD can also perform well on prediction tasks. To the best of our knowledge, we are the first to address regression and classification tasks with a generative model efficiently. More experimental details and additional experimental results can be found in Appendix C.

### 6.1 Generation task

For both unconditional and conditional generation, we use QM9 [Ramakrishnan et al., 2014], one of the most widely adopted datasets in molecular machine learning research, which is suitable for both generation and regression tasks. QM9 contains both graph and 3D structures together with several quantum properties for 130k small molecules, limited to 9 heavy atoms. We provide more generation results on larger dataset MOSES [Polykovskiy et al., 2020] in Appendix C.

Table 1: Unconditional generation results on QM9.

| Model | Validity(%)↑ | Uniqueness(%)↑ | FCD↓ | NSPDK↓ | Novelty(%)↑ |
|---|---|---|---|---|---|
| MoFlow | 91.36 | **98.65** | 4.47 | 0.017 | 94.72 |
| GraphAF | 74.43 | 88.64 | 5.27 | 0.020 | 86.59 |
| GraphDF | 93.88 | 98.58 | 10.93 | 0.064 | **98.54** |
| GDSS | 95.72 | 98.46 | 2.9 | 0.003 | 86.27 |
| DiGress | 99.01 | 96.34 | 0.25 | 0.0003 | 35.46 |
| HGGT | 99.22 | 95.65 | 0.40 | 0.0003 | 24.01 |
| GruM | **99.69** | 96.90 | 0.11 | **0.0002** | 24.15 |
| LGD-small (ours) | 98.46 | 97.53 | 0.32 | 0.0004 | 56.35 |
| LGD-large (ours) | 99.13 | 96.82 | **0.10** | **0.0002** | 38.56 |

Table 2: Conditional generation results on QM9 (MAE ↓)

| Model | $\mu$ | $\alpha$ | $\epsilon_{HOMO}$ | $\epsilon_{LUMO}$ | $\Delta\epsilon$ | $c_v$ |
|---|---|---|---|---|---|---|
| $\omega$ [Xu et al., 2023] | 0.043 | 0.10 | 39 | 36 | 64 | 0.040 |
| $\omega$ (ours) | 0.058 | 0.06 | 18 | 24 | 28 | 0.038 |
| Random | 1.616 | 9.01 | 645 | 1457 | 1470 | 6.857 |
| $N_{atom}$ | 1.053 | 3.86 | 426 | 813 | 866 | 1.971 |
| EDM | 1.111 | 2.76 | 356 | 584 | 655 | 1.101 |
| GeoLDM | 1.108 | **2.37** | 340 | **522** | 587 | 1.025 |
| LGD (ours) | **0.879** | 2.43 | **313** | 641 | **586** | **1.002** |

**Unconditional generation.** In the unconditional molecular modeling and generation task, we measure the model's ability to learn the distribution of molecular data and generate molecules that are both chemically valid and structurally diverse. Following the common setting [Xu et al., 2023], we generate 10k molecules in the evaluation stage, and report the validity and uniqueness, which are the percentages of valid (measured by RDKIT) and unique molecules among the generated molecules. We also report neighborhood subgraph pairwise distance kernel (NSPDK) and Frechet ChemNet Distance (FCD) metrics to measure the similarity between generated samples and the test set. We leave further discussions on relaxed validity [Jo et al., 2022] and novelty in Appendix C. For baseline models, we select classical and recent strong models including MoFlow [Zang and Wang, 2020], GraphAF [Shi* et al., 2020], GraphDF [Luo et al., 2021], GDSS [Jo et al., 2022], DiGress [Vignac et al., 2022], HGGT [Jang et al., 2024], GruM [Jo et al., 2024].

The results are shown in Table 1. LGD-large achieves the best results in FCD and NSPDK metrics, and is highly competitive in validity and uniqueness. This verifies the advantages of our latent diffusion model and the ability to generate nodes and edges simultaneously. It is worth mentioning that our LGD works in a different and potentially more difficult setting compared with some works like GeoLDM [Xu et al., 2023]. In particular, LGD generates both atoms (nodes) and bonds (edges) simultaneously in one shot, while GeoLDM only generates atoms (nodes), and then computes bonds (edges) with a strong decoder based on pair-wise atomic distances and atom types. We also do not use any 3D information, so it is indirect to compare with 3D baselines due to different training settings.

**Conditional generation.** We still use the QM9 dataset for our conditional generation task, where we aim to generate target molecules with the desired chemical properties. We follow the settings of [Xu et al., 2023] and split the training set into two parts, each containing 50k molecules. We train the latent diffusion model and separate property prediction networks $\omega$ (with architectures described in Equation (58)) on two halves, respectively. For evaluation, we first generate samples using the latent diffusion model given the conditioning property $y$, and use $\omega$ to predict the property $\hat{y}$ of the generated molecules. We measure the Mean Absolute Error between $y$ and $\hat{y}$, and experiment with six properties: Dipole moment $\mu$, polarizability $\alpha$, orbital energies $\epsilon_{HOMO}, \epsilon_{LUMO}$ and their gap $\Delta\epsilon$, and heat capacity $C_v$.

We report EDM [Hoogeboom et al., 2022] and GeoLDM [Xu et al., 2023] as baseline models. We also incorporate (a) the MAE of the regression model $\omega$ of ours and [Xu et al., 2023], which can be viewed as a lower bound of the generative models; (b) *Random*, which shuffle the labels and evaluate $\omega$, thus can be viewed as the upper bound of the MAE metric; (c) $N_{\text{atoms}}$, which predicts the properties based only on the number of atoms in the molecule.

As shown in Table 2, even if we do not use 3D information and generate both atoms and bonds in one simultaneously, LGD achieves the best results in 4 out of 6 properties. The results verify the excellent controllable generation capacity of LGD.

## 6.2 Prediction with conditional generative models

We evaluate LGD extensively on prediction tasks, including regression and classification tasks of different levels. More results can be found in Appendix C.

**Regression.** For regression task, we select ZINC12k [Dwivedi et al., 2020], which is a subset of ZINC250k containing 12k molecules. The task is molecular property (constrained solubility) regression, measured by MAE. We use the official split of the dataset.

Note that we are the first to use generative models to perform regression tasks, therefore no comparable generative models can be selected as baselines. Therefore, we choose traditional regression models, including GIN [Xu et al., 2018], PNA [Corso et al., 2020], DeepLRP [Chen et al., 2020], OSAN [Qian et al., 2022], KP-GIN+ [Feng et al., 2022a], GNN-AK+ [Zhao et al., 2021], CIN [Bodnar et al., 2021] and GPS [Rampásek et al., 2022] for comparison.

We train our LGD model and test inference with both DDPM and DDIM methods. While the generated predictions are not deterministic, we only predict once for each graph for a fair comparison with deterministic regression

Table 3: Zinc12K results (MAE $\downarrow$). Shown is the mean $\pm$ std of 5 runs.

| Method | Test MAE |
|---|---|
| GIN | $0.163 \pm 0.004$ |
| PNA | $0.188 \pm 0.004$ |
| GSN | $0.115 \pm 0.012$ |
| DeepLRP | $0.223 \pm 0.008$ |
| OSAN | $0.187 \pm 0.004$ |
| KP-GIN+ | $0.119 \pm 0.002$ |
| GNN-AK+ | $0.080 \pm 0.001$ |
| CIN | $0.079 \pm 0.006$ |
| GPS | $0.070 \pm 0.004$ |
| LGD-DDIM (ours) | $0.081 \pm 0.006$ |
| LGD-DDPM (ours) | $\mathbf{0.065} \pm 0.003$ |

models. We will show in Appendix C that ensemble techniques can further improve the quality of prediction. As shown in Table 3, LDM (with DDPM) achieves the best results, even outperforming the powerful graph transformers GPS [Rampásek et al., 2022]. In comparison, the regression model with the same graph attention architecture as the score network in LGD can only achieve a worse $0.084 \pm 0.004$ test MAE, which validates the advantage of latent diffusion model over traditional regression models. We also observe that inference with DDIM is much faster, but may lead to a performance drop, which aligns with previous observations [Song et al., 2020a, Cao et al., 2023]. Moreover, our LGD requires much less training steps compared with GraphGPS, see Appendix C for more detailed discussions on experimental findings.

**Classification.** We choose node-level tasks for classification. Datasets include co-purchase graphs from Amazon (Photo) [Shchur et al., 2018], coauthor graphs from Coauthor (Physics) [Shchur et al., 2018], and the citation graph OGBN-Arxiv with over 169K nodes. The common $60\%, 20\%, 20\%$ random split is adopted for Photo and Physics, and the official split based on publication dates of the papers is adopted for OGBG-Arxiv.

We choose both classical GNN models and state-of-the-art graph transformers as baselines. GNNs include GCN [Kipf and Welling, 2016], GAT [Velickovic et al., 2017], GraphSAINT [Zeng et al., 2020], GRAND+ [Feng et al., 2022b]. Graph transformers include Graphormer [Ying et al., 2021], SAN [Kreuzer et al., 2021], GraphGPS [Rampásek et al., 2022], and the scalable Exphormer [Shirzad et al., 2023] and NAGphormer [Chen et al., 2023a].

As reported in Table 4, our LGD not only scales to these datasets while a number of complex models like GraphGPS Rampásek et al. [2022] fail to do so, but also achieves the best results, even outperforming those state-of-the-art graph transformers including Exphormer [Shirzad et al., 2023] and NAGphormer [Chen et al., 2023a]. Overall, our LGD reveals great advantages in both scalability and task performance.

Table 4: Node-level classification tasks (accuracy ↑) on datasets from Amazon, Coauthor and OGBN-Arxiv. Reported are mean $\pm$ std over 10 runs with different random seeds. Highlighted are **best** results.

| Dataset | Photo | Physics | OGBN-Arxiv |
|---|---|---|---|
| GCN | $92.70 \pm 0.20$ | $96.18 \pm 0.07$ | $71.74 \pm 0.29$ |
| GAT | $93.87 \pm 0.11$ | $96.17 \pm 0.08$ | - |
| GraphSAINT | $91.72 \pm 0.13$ | $96.43 \pm 0.05$ | - |
| GRAND+ | $94.75 \pm 0.12$ | $96.47 \pm 0.04$ | - |
| Graphormer | $92.74 \pm 0.13$ | OOM | OOM |
| SAN | $94.86 \pm 0.10$ | OOM | OOM |
| GraphGPS | $95.06 \pm 0.13$ | OOM | OOM |
| Exphormer | $95.35 \pm 0.22$ | $96.89 \pm 0.09$ | $72.44 \pm 0.28$ |
| NAGphormer | $95.49 \pm 0.11$ | $97.34 \pm 0.03$ | - |
| LGD (ours) | $\mathbf{96.94 \pm 0.14}$ | $\mathbf{98.55 \pm 0.12}$ | $\mathbf{73.17 \pm 0.22}$ |

## 7 Conclusions and Limitations

We propose Latent Graph Diffusion (LGD), the first graph generative framework that is capable of solving tasks of all types (generation, regression and classification) and all levels (node, edge, and graph). We conceptually formulate regression and classification tasks as conditional generation, and show that latent diffusion models can complete them with provable theoretical guarantees. We then encode the graph into a powerful latent space and train a latent diffusion model to generate graph representations with high qualities. Leveraging specially designed graph transformer architectures and cross-attention mechanisms, LGD can generate node, edge, and graph features simultaneously in both unconditional or conditional settings. We experimentally show that LGD is not only capable of completing these tasks of different types and levels, but also capable of achieving extremely competitive performance. We believe that our work is a solid step towards graph foundation models, providing fundamental architectures and theoretical guarantees. We hope it could inspire more extensive future research.

There are still some limitations of this paper that could be considered as future work. First, although the LGD is a unified framework, we train models separately for each task in our experiments. To build a literal foundation model, we need to train a single model that can handle different datasets, even from different domains. This requires expensive computation resources, more engineering techniques and extensive experiments. Second, it would be meaningful to verify the effectiveness of utilizing diffusion to perform deterministic prediction tasks in other domains, such as computer vision.

## Acknowledgments and Disclosure of Funding

Muhan Zhang is supported by the National Natural Science Foundation of China (62276003).

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

# A Background on Diffusion Models

In this section we provide more background knowledge on diffusion models, including formal formulation of the families of DDPM [Ho et al., 2020] and SMLD [Song and Ermon, 2019] methods, as well as their unified formulation through the framework of SDE [Song et al., 2020b] or ODE [Song et al., 2020a].

## A.1 Denoising diffusion probabilistic models

Given samples from the ground truth data distribution $q(\mathbf{x}_0)$, generative models aim to learn a model distribution $p_\theta(\mathbf{x}_0)$ that approximates $q(\mathbf{x}_0)$ well and is easy to sample from. Denoising dffusion probabilistic models (DDPMs) [Ho et al., 2020] are a family of latent variable models of the form

$$p_\theta(\mathbf{x}_0) = \int p_\theta(\mathbf{x}_{0:T}) \mathrm{d}\mathbf{x}_{1:T} \tag{10}$$

where

$$p_\theta(\mathbf{x}_{0:T}) := p_\theta(\mathbf{x}_T) \prod_{t=1}^{T} p_\theta^{(t)}(\mathbf{x}_{t-1}|\mathbf{x}_t) \tag{11}$$

here $\mathbf{x}_1, \ldots, \mathbf{x}_T$ are latent variables in the same sample space (denoted as $\mathcal{X}$) as the data $\mathbf{x}_0$.

A special property of diffusion models is that the approximate posterior $q(\mathbf{x}_{1:T}|\mathbf{x}_0)$, also called the *forward process* or *diffusion process*, is a fixed Markov chain which gradually adds Gaussian noise to the data according to a variance schedule $\beta_1, \ldots, \beta_T$:

$$q(\mathbf{x}_{1:T}|\mathbf{x}_0) := \prod_{t=1}^{T} q(\mathbf{x}_t|\mathbf{x}_{t-1}) \tag{12}$$

where

$$q(\mathbf{x}_t|\mathbf{x}_{t-1}) := \mathcal{N}(\mathbf{x}_t; \sqrt{1-\beta_t}\mathbf{x}_{t-1}, \beta\mathbf{I}) \tag{13}$$

The forward process variance $\beta_t$ can either be held constant as hyperparameters, or learned by reparameterization [Ho et al., 2020]. One advantage of the above parameterization is that sampling $\mathbf{x}_t$ at an arbitrary timestep $t$ has a closed form,

$$q(\mathbf{x}_t|\mathbf{x}_0) = \mathcal{N}(\mathbf{x}_t; \sqrt{\bar{\alpha}_t}\mathbf{x}_0, (1-\bar{\alpha}_t)\mathbf{I}) \tag{14}$$

where $\alpha_t := 1 - \beta_t$ and $\bar{\alpha}_t := \prod_{s=1}^{t} \alpha_s$. Utilizing the closed-form expression, we can express $\mathbf{x}_t$ as a linear combination of $\mathbf{x}_0$ and a noise variable $\epsilon \sim \mathcal{N}(\mathbf{0}, \mathbf{I})$:

$$\mathbf{x}_t = \sqrt{\bar{\alpha}_t}\mathbf{x}_0 + \sqrt{1-\bar{\alpha}_t}\epsilon \tag{15}$$

Note that $\bar{\alpha}_t$ is a decreasing sequence, and when we set $\bar{\alpha}_T$ sufficiently close to 0, $q(\mathbf{x}_T|\mathbf{x}_0)$ converges to a standard Gaussian for all $\mathbf{x}_0$, thus we can naturally set $p_\theta(\mathbf{x}_T) := \mathcal{N}(\mathbf{0}, \mathbf{I})$.

In the *generative process*, the model parameterized by $\theta$ are trained to fit the data distribution $q(\mathbf{x}_0)$ by approximating the intractable *reverse process* $q(\mathbf{x}_{t-1}|\mathbf{x}_t)$. This can be achieved by maximizing a variational lower bound:

$$\max_\theta \mathbb{E}_{q(\mathbf{x}_0)}[\log p_\theta(\mathbf{x}_0)] \leq \max_\theta \mathbb{E}_{q(\mathbf{x}_0, \mathbf{x}_1, \ldots, \mathbf{x}_T)}[\log p_\theta(\mathbf{x}_{0:T}) - \log q(\mathbf{x}_{1:T}|\mathbf{x}_0)] \tag{16}$$

The expressivity of the reverse process is ensured in part by the choice of conditionals in $p_\theta(\mathbf{x}_{t-1}|\mathbf{x}_t)$. If all the conditionals are modeled as Gaussians with trainable mean functions and fixed variances as in [Ho et al., 2020], then the objective in Equation (16) can be simplified to

$$\mathcal{L}_\gamma(\epsilon_\theta) := \sum_{t=1}^{T} \lambda_t \mathbb{E}_{\mathbf{x}_0 \sim q(\mathbf{x}_0), \epsilon_t \sim \mathcal{N}(\mathbf{0}, \mathbf{I})} \left[ ||\epsilon_\theta^{(t)}(\sqrt{\bar{\alpha}_t}\mathbf{x}_0 + \sqrt{1-\bar{\alpha}_t}\epsilon_t) - \epsilon_t||_2^2 \right] \tag{17}$$

where $\epsilon_\theta := \{\epsilon_\theta^{(t)}\}_{t=1}^{T}$ is a set of $T$ number of functions, each $\epsilon_\theta^{(t)} : \mathcal{X} \to \mathcal{X}$ indexed by $t$ is a denoising function with trainable parameters $\theta^{(t)}$; $\lambda := [\lambda_1, \ldots, \lambda_T]$ is a positive coefficients, which is set to all ones in [Ho et al., 2020] and [Song and Ermon, 2019], and we follow their settings throughout the paper.

In the inference stage for a trained model, $\mathbf{x}_0$ is sampled by first sampling $\mathbf{x}_T$ from the prior $p_\theta(\mathbf{x}_T)$, followed by sampling $\mathbf{x}_{t-1}$ from the above generative processes iteratively. The length of steps $T$ in the forward process is a hyperparameter (typically $T = 1000$) in DDPMs. While large $T$ enables the Gaussian conditionals distributions as good approximations, the sequential sampling becomes slow. DDIM [Song et al., 2020a] proposes to sample from generalized generative process as follows,

$$\mathbf{x}_{t-1} = \sqrt{\bar{\alpha}_{t-1}} \left( \frac{\mathbf{x}_t - \sqrt{1 - \bar{\alpha}_t} \epsilon_\theta^{(t)}(\mathbf{x}_t)}{\sqrt{\bar{\alpha}_t}} \right) + \sqrt{1 - \bar{\alpha}_{t-1} - \sigma_t^2} \cdot \epsilon_\theta^{(t)}(\mathbf{x}_t) + \sigma_t \epsilon_t \tag{18}$$

When $\sigma_t = \sqrt{(1 - \bar{\alpha}_{t-1})(1 - \alpha_t/\alpha_{t-1})/(1 - \alpha_t)}$ for all $t$, the generative process becomes a DDPM, while $\sigma_t = 0$ results in DDIM, which is an implicit probabilistic model that samples with a fixed procedure, thus corresponding to ODE [Song et al., 2020a]. In our work, both DDPM and DDIM are considered in our generative process.

## A.2  Score matching with Langevin dynamics

Although we mainly illustrate our framework using the family of DDPM and DDIM in the main text, the family of score matching with Langevin dynamics (SMLD) [Song and Ermon, 2019] methods are also applicable in our LGD.

Score matching methods estimates the score (gradient of log-likelihood) of the data using a score network $s_\theta : \mathbb{R}^d \to \mathbb{R}^d$ by minimizing the objective

$$\mathbb{E}_{q(\mathbf{x})} \left[ ||s_\theta(\mathbf{x}) - \nabla_\mathbf{x} \log q(\mathbf{x})||_2^2 \right] \tag{19}$$

Given the equivalence of SMLD and DDPM explained in Appendix A.3, we do not distinguish the two families of generative models in most cases of the paper. We also sometimes use the notation $\epsilon_\theta$ to represent both the denoising network in DDPM and the score network in SMLD.

## A.3  Unified formulation through the framework of SDE and ODE

Song et al. [2020b] formulates both SMLD and DDPM through the system of SDE. The authors further propose ODE samplers which are deterministic compared with SDE, so does [Song et al., 2020a]. Liu et al. [2022] use ODE-based rectified flow to sample with high efficiency. Cao et al. [2023] and Deveney et al. [2023] analyze the connections and comparisons between ODE and SDE based methods.

In particular, the diffusion process can be modeled as the solution to an SDE:

$$\mathrm{d}\mathbf{x} = \mathbf{f}(\mathbf{x}, t)\mathrm{d}t + g(t)\mathrm{d}\mathbf{w} \tag{20}$$

where $\mathbf{w}$ is the standard Wiener process (a.k.a., Brownian motion), $\mathbf{f}(\cdot, t) : \mathbb{R}^d \to \mathbb{R}^d$ is a vector-valued function called the *drift* coefficient of $\mathbf{x}(t)$, and $g(\cdot) : \mathbb{R} \to \mathbb{R}$ is a scalar function called the *diffusion* coefficient of $\mathbf{x}(t)$. Note that the end time $T$ of the diffusion process described in the SDE has a different meaning in the number of discrete diffusion steps in DDPM, and we denote the latter as $N \in \mathbb{N}$ in the SDE formulations.

To sample from the backward process, we consider the reverse-time SDE which is also a diffusion process,

$$\mathrm{d}\mathbf{x} = [\mathbf{f}(\mathbf{x}, t) - g(t)^2 \nabla_\mathbf{x} \log p_t(\mathbf{x})]\mathrm{d}t + g(t)\mathrm{d}\bar{\mathbf{w}} \tag{21}$$

where $\bar{\mathbf{w}}$ is a standard Wiener process in which time reverse from $T$ to $0$. Again, here $T$ is the end time of the diffusion process, which is different from the number of discrete diffusion steps in DDPM). $\mathrm{d}t$ is an infinitesimal negative timestep, which is discretized in implementation of DDPM, i.e. each time step is $h := T/N$ where $N$ is the number of discrete steps in DDPM.

## A.4  Related work on theoretical analysis of diffusion models

The empirical success of diffusion models aroused researchers' interest in theoretical analysis of them recently. Chen et al. [2023b] and Yuan et al. [2023] provide some theoretical analysis on score approximation and reward-directed conditional generation, respectively. Li et al. [2023b] provide generalization bound of diffusion models. Li et al. [2023a] provide empirical evidence that diffusion

models can be zero-shot classifiers. Benton et al. [2023] analyzes the error bound of flow matching methods, and Lee et al. [2022], Chen et al. [2022] analyzed the convergence of SDE-based score matching methods.

# B  Theoretical Analysis

In this section, we provide our novel theoretical analysis on the guarantees of solving regression or classification tasks with a conditional latent diffusion model in Section 5. Our theory is generic and is not limited to graph data.

As described in the main text, while training the conditional latent diffusion for prediction tasks with labels, we aim to generate the latent embedding of labels $y$ conditioning on the latent embedding of data $x$. We first consider regression tasks in our analysis, where the label is a scaler $y$; classification tasks can be analyzed in a similar manner. We use a slightly different notation compared with the original DDPM formulation in [Ho et al., 2020]; instead, we use $N$ to denote the maximum number of discrete diffusion steps in a DDPM (which is denoted as $T$ in [Ho et al., 2020]), and instead use $T$ to denote the end time in the SDE which controls the diffusion process.

According to the reasons in Section 5, we choose to model the joint distribution of $x, y$, thus use $\tau(x, y) \in \mathbb{R}^d$ to jointly encode the data and the label, and use $\mathcal{E}(x) \in \mathbb{R}^d$ to embed the data (condition) only. Here we suppose the output of each encoder is a $d$-dimensional vector. The diffusion model is trained to generate $\tau(x, y)$ conditioning on $\mathcal{E}(x)$. We suppose two encoders share one decoder $\mathcal{D}_w$ which is linear layer parameterized by $w \in \mathbb{R}^d$. First we have the assumptions on the reconstruction errors of the autoencodings.

**Assumption B.1.** (First and second moment bound of autoencoding errors.)

$$\epsilon_1 := \mathbb{E}_q[||w^\top \tau(x, y) - y||] < \infty \tag{22}$$

$$m_1^2 := \mathbb{E}_q[||w^\top \tau(x, y) - y||^2] < \infty \tag{23}$$

$$\epsilon_2 := \mathbb{E}_q[||w^\top \mathcal{E}(x) - y||] < \infty \tag{24}$$

$$m_2^2 := \mathbb{E}_q[||w^\top \mathcal{E}(x) - y||^2] < \infty \tag{25}$$

Typically, $\epsilon_2$ is just the MAE of a traditional deterministic regression model; $\epsilon_1 \ll \epsilon_2$ since $\tau$ takes $y$ as inputs and thus has an extremely small reconstruction error. Assumption B.1 generally holds and can be verified across all experiments.

To show why a generative model can perform better than the regression model, i.e. could have a smaller error than $\epsilon_2$, we continue to make the following mild assumptions on the data distribution $q$, which also generally holds for practical data..

**Assumption B.2.** (Lipschitz score). For all $t \geq 0$, the score $\nabla \ln q_t$ is $L$-Lipschitz.

Finally, the quality of diffusion obviously depends on the expressivity of score matching network (denoising network) $\epsilon_\theta^{(t)}$.

**Assumption B.3.** (Score estimation error). For all $k = 1, \ldots, N$,

$$\mathbb{E}_{q_{kh}}[\epsilon_\theta^{kh} - \nabla \ln q_{kh}] \leq \epsilon_{\text{score}}^2 \tag{26}$$

where $h := T/N$ is the time step in the SDE.

This is the same assumption as in [Chen et al., 2022]. In our main paper the denoising network is a powerful graph transformer with architectures described in Equation (58), which can well guarantee a small score estimation error. However, detailed bounds of $\epsilon_{\text{score}}$ is beyond the scope of this paper.

Finally, we consider a simple version of embedding the condition, where the denoising network $\epsilon_\theta$ still samples in the same way as unconditional generation, except that we directly add the embedding of condition to the final output of $\epsilon_\theta$. In other words,

$$\epsilon_\theta^{(t)}(z, \mathcal{E}(x)) = \epsilon_\theta^{(t)}(z) + \mathcal{E}(x) \tag{27}$$

where $z$ is the latent being sampled.

Now we give our main theorem of the regression MAE of the latent diffusion. The theorem is proved through the SDE description of DDPM as in Appendix A.3, where we use $T$ to denote the end time of diffusion time in the SDE, $N$ the number of discrete diffusion steps in DDPM implementation, and $\gamma^d := \mathcal{N}(\mathbf{0}, \mathbb{I}_d)$ as a simplified notation of a random variable from the standard Gaussian distribution.

**Theorem B.4.** *Suppose Assumption B.1, Assumption B.2, Assumption B.3 hold, and the step size $h := T/N$ satisfies $h \preceq 1/L$ where $L \geq 1$. Then the mean absolute error (MAE) of the conditional latent diffusion model in the regression task is bounded by*

$$\text{MAE} \leq \mathbb{E}_q[||w^\top \hat{\tau}(\boldsymbol{x}, y) - w^\top \tau(\boldsymbol{x}, y)||] + \epsilon_1 \tag{28}$$

$$\preceq \underbrace{\sqrt{\text{KL}(q_z || \gamma^d)} \exp(-T)}_{\text{convergence of forward process}} + \underbrace{(L\sqrt{dh} + L(m_1 + m_2)h)\sqrt{T}}_{\text{discretization error}} + \underbrace{\epsilon_{\text{score}}\sqrt{T}}_{\text{score estimation error}} + \underbrace{\epsilon_1}_{\text{encoder error}} \tag{29}$$

*where $q_z$ is the ground truth distribution of $\boldsymbol{z} := w^t\big(\tau(\boldsymbol{x}, y) - \mathcal{E}(\boldsymbol{x})\big)$.*

*Proof.* Conditioning on $\mathcal{E}(\boldsymbol{x})$, the diffusion model $\epsilon_\theta$ aims to generate the refined representation $\tau(\boldsymbol{x}, y)$. Denote the generated embedding as $\hat{\tau}(\boldsymbol{x}, y)$, then

$$\text{MAE} := \mathbb{E}_q[||w^\top \hat{\tau}(\boldsymbol{x}, y) - y||] \tag{30}$$

$$= \mathbb{E}_q[||(w^\top \hat{\tau}(\boldsymbol{x}, y) - w^\top \tau(\boldsymbol{x}, y)) + (w^\top \tau(\boldsymbol{x}, y) - y)||] \tag{31}$$

$$\leq \mathbb{E}_q[||w^\top \hat{\tau}(\boldsymbol{x}, y) - w^\top \tau(\boldsymbol{x}, y)||] + \mathbb{E}_q[||w^\top \tau(\boldsymbol{x}, y) - y||] \tag{32}$$

$$= \mathbb{E}_q[||w^\top \hat{\tau}(\boldsymbol{x}, y) - w^\top \tau(\boldsymbol{x}, y)||] + \epsilon_1 \tag{33}$$

We only need to prove the bound of $\mathbb{E}_q[||w^\top \hat{\tau}(\boldsymbol{x}, y) - w^\top \tau(\boldsymbol{x}, y)||]$. Note that the generation is conditioned on $\mathcal{E}(\boldsymbol{x})$. Instead of cross attention, we consider a simple method to incorporate the condition, which use a residual connection that directly add the condition $\mathcal{E}(\boldsymbol{x})$ to the feature being denoised. Then the conditional generation is equivalent to the unconditional generation which samples the distribution of $\tau(\boldsymbol{x}, y) - \mathcal{E}(\boldsymbol{x})$ from noise. It suffice to show the unconditional generation error of $\tau(\boldsymbol{x}, y) - \mathcal{E}(\boldsymbol{x})$. Using the notation

$$\boldsymbol{z}_1 := w^\top \tau(\boldsymbol{x}, y) - y \tag{34}$$

$$\boldsymbol{z}_2 := w^\top \mathcal{E}(\boldsymbol{x}) - y \tag{35}$$

where

$$\mathbb{E}_q[||\boldsymbol{z}_1||] = \epsilon_1 \tag{36}$$

$$\mathbb{E}_q[||\boldsymbol{z}_1||^2] = m_1^2 \tag{37}$$

$$\mathbb{E}_q[||\boldsymbol{z}_2||] = \epsilon_2 \tag{38}$$

$$\mathbb{E}_q[||\boldsymbol{z}_2||^2] = m_2^2 \tag{39}$$

We then have

$$\boldsymbol{z} := w^\top\big(\tau(\boldsymbol{x}, y) - \mathcal{E}(\boldsymbol{x})\big) = \boldsymbol{z}_1 - \boldsymbol{z}_2 \tag{40}$$

To complete our derivation, consider the following lemma proposed in [Chen et al., 2022],

**Lemma B.5.** (TV distance of DDPM) [Chen et al., 2022]. *Suppose that Assumption B.1, Assumption B.2, Assumption B.3 hold. The data to be generated $\boldsymbol{z}$ has the ground truth ground truth distribution $q_z$ and the estimated distribution by the diffusion model is $p_z$. The second moment bond $M^2 := \mathbb{E}_{q_z}[|| \cdot ||^2] > \infty$. Suppose that the step size $h := T/N$ satisfies $h \preceq 1/L$ where $L \geq 1$. Then the total variance (TV) distance satistfies*

$$\text{TV}(p_z, q_z) \preceq \sqrt{\text{KL}(q_z || \gamma^d)} \exp(-T) + (L\sqrt{dh} + LMh)\sqrt{T} + \epsilon_{\text{score}}\sqrt{T} \tag{41}$$

The proof of the lemma is in [Chen et al., 2022].

Back to our derivation, utilizing the Cauchy inequality, we have

$$\mathbb{E}_{q_z}[||\boldsymbol{z}||^2] = \mathbb{E}_{q_z}[||\boldsymbol{z}_1 - \boldsymbol{z}_2||^2] \tag{42}$$

$$\leq \mathbb{E}_{q_z}[||\boldsymbol{z}_1||^2] + \mathbb{E}_{q_z}[||\boldsymbol{z}_2||^2] + 2\mathbb{E}_{q_z}[||\boldsymbol{z}_1\boldsymbol{z}_2||] \tag{43}$$

$$\leq m_1^2 + m_2^2 + 2m_1 m_2 < \infty \tag{44}$$

thus

$$\mathrm{TV}(p_z, q_z) := \frac{1}{2}\mathbb{E}_{q_z}||\hat{\boldsymbol{z}} - \boldsymbol{z}|| \preceq \sqrt{\mathrm{KL}(q_z||\gamma^d)}\exp(-T) + (L\sqrt{dh} + L(m_1 + m_2)h)\sqrt{T} + \epsilon_{\mathrm{score}}\sqrt{T} \tag{45}$$

where $\hat{\boldsymbol{z}} \sim p_z$ is the estimated embedding of $\boldsymbol{z}$ generated by the diffusion model. Recall the condition embedding is implemented via simply adding the condition to the generated $\hat{\boldsymbol{z}}$, i.e. $w^\top \hat{\tau}(\boldsymbol{x}, y) = \hat{\boldsymbol{z}} + w^\top \mathcal{E}(\boldsymbol{x})$, we have

$$\mathbb{E}_q[||w^\top \hat{\tau}(\boldsymbol{x}, y) - w^\top \tau(\boldsymbol{x}, y)||] \tag{46}$$

$$=\mathbb{E}_q[||\hat{\boldsymbol{z}} - (w^\top \tau(\boldsymbol{x}, y) - w^\top \mathcal{E}(\boldsymbol{x}))||] \tag{47}$$

$$=\mathbb{E}_{q_z}[||\hat{\boldsymbol{z}} - \boldsymbol{z}||] \tag{48}$$

$$\preceq\sqrt{\mathrm{KL}(q_z||\gamma^d)}\exp(-T) + (L\sqrt{dh} + L(m_1 + m_2)h)\sqrt{T} + \epsilon_{\mathrm{score}}\sqrt{T} \tag{49}$$

where we omit the constant factor 2 in Equation (45). Finally we complete our proof,

$$\mathrm{MAE} \leq \mathbb{E}_q[||w^\top \hat{\tau}(\boldsymbol{x}, y) - w^\top \tau(\boldsymbol{x}, y)||] + \epsilon_1 \tag{50}$$

$$\preceq \underbrace{\sqrt{\mathrm{KL}(q_z||\gamma^d)}\exp(-T)}_{\text{convergence of forward process}} + \underbrace{(L\sqrt{dh} + L(m_1 + m_2)h)\sqrt{T}}_{\text{discretization error}} + \underbrace{\epsilon_{\mathrm{score}}\sqrt{T}}_{\text{score estimation error}} + \underbrace{\epsilon_1}_{\text{encoder error}} \tag{51}$$

$$\square$$

We now give more interpretations to Theorem B.4. The first term $\sqrt{\mathrm{KL}(q_z||\gamma^d)}\exp(-T)$ is the error of convergence of the forward process, which can be exponetially small when we have enough SDE diffusion time $T$, or equivalently, $\bar{\alpha}$ is efficiently small to make $q_T$ close enough to the standard Gaussian $\gamma^d := \mathcal{N}(\boldsymbol{0}, \mathbf{I}_d)$.

The second term $(L\sqrt{dh} + L(m_1 + m_2)h)\sqrt{T}$ is the discretization error, since when implementing the DDPM we have to use discrete $N$ steps to simulate the ODE. It is interesting to see that this term is affected by both the latent dimension $d$ and the second moment of the variation encoding errors $m_1$ and $m_2$ (i.e. the reconstruction MSE loss of labels and conditions). The results implies that a lower dimension of the latent space $\mathcal{H}$ is more suitable for learning a diffusion model. However, note that the reconstruction MSE loss $m_1, m_2$ may also be affected by $d$: when the rank of data is high, a low dimension latent space may not represent the data well, and the projection made by the decoder $w \in \mathbb{R}^d$ is likely to cause information loss, thus increasing $m_1$ and $m_2$. Therefore, the latent dimension $d$ should be carefully chosen, which should not be neither too low (poor representation power and large reconstruction error) nor too high (large diffusion error). This verifies the intuition discussed in Section 4.

The third term is the score estimation error, which is related to the expressivity of denoising (score) networks. Detailed theoretical analysis of various architectures is beyond the scope of the paper, and we refer readers to more relevant papers on expressivity or practical performance of potential architectures, including MPNN [Xu et al., 2018], high-order and subgraph GNNs [Morris et al., 2019, 2020, Feng et al., 2022a, Zhou et al., 2023b, Zhang et al., 2023a], and graph transformers [Kim et al., 2022, Zhou et al., 2024]. For prediction tasks where we have ground truth graph structures as inputs, we can also use positional encodings and structural encodings to enhance the theoretical and empirical expressivity of score networks [Zhang et al., 2023b, Dwivedi et al., 2021, Wang et al., 2022, Zhou et al., 2023a, Lim et al., 2022]. It is remarkable that our architecture reveals strong performance in experiments. Empirically, the graph transformer in Equation (58) we adopted is powerful in function

approximation. In the language of distinguishing non-isomorphic graphs, our transformer is at least as powerful as 1-order Weisfeiler-Lehman test [Xu et al., 2018, Zhou et al., 2024].

The last term is the error brought by autoencoding. However, as $\epsilon_1$ is the reconstruction MAE of $\tau$ which aims to encode the labels, $\epsilon_1$ is typically much smaller than the MAE of regression models $\epsilon_2$. Our experimental observations support the above assumption. Therefore, **the conditional generation model outperforms the traditional regression model** as long as

$$\sqrt{\mathrm{KL}(q_z||\gamma^d)}\exp(-T) + (L\sqrt{dh} + L(m_1+m_2)h)\sqrt{T} + \epsilon_{\mathrm{score}}\sqrt{T} + \epsilon_1 \preceq \epsilon_2 \qquad (52)$$

This can be achieved by searching hyperparameters of the diffusion model, including $T, N, h$. Intuitively, a sufficiently small $h$ (or equivalently, sufficiently large number of diffusion steps $N$) would lead to a small error. Following [Chen et al., 2022], suppose $\mathrm{KL}(q_z||\gamma^d)\exp(-T) \le \mathrm{poly}(d)$, $m_1 + m_2 \le d$, then by choosing $T \asymp \ln(\mathrm{KL}(q_z||\gamma^d)/\epsilon)$ and $h \asymp \frac{\epsilon^2}{L^2 d}$ and hiding logarithmic factors, we have

$$\mathrm{MAE} \le O(\epsilon + \epsilon_{\mathrm{score}} + \epsilon_1) \qquad (53)$$

where $\epsilon$ is the error scale we desire. In this case, we need $N = \Theta(\frac{L^2 d}{\epsilon^2})$ number of discrete diffusion steps. If $\epsilon_{\mathrm{score}} \le O(\epsilon)$ and $\epsilon_1 \le O(\epsilon)$, then $\mathrm{MAE} \le \epsilon$.

Finally, we have the following statement which is straightforward to prove.

**Corollary B.6.** *When the condition is embedded as in Equation* (27)*, there exists at least one latent diffusion model whose MAE is not larger than the autoencoder.*

*Proof.* The extreme case is that the denoising network always outputs $\epsilon_\theta^{(t)}(\boldsymbol{z}) \equiv 0$ for all $t$, then according to Equation (27), the generated representation

$$\hat{\tau}(\boldsymbol{x}, y) \equiv \mathcal{E}(\boldsymbol{x}) \qquad (54)$$

Then the MAE would be simply the MAE of the autoencoder:

$$\mathrm{MAE} = \mathbb{E}_q[||w^T\hat{\tau}(\boldsymbol{x}, y) - y||] \qquad (55)$$
$$= \mathbb{E}_q[||w^T\mathcal{E}(\boldsymbol{x}) - y||] \qquad (56)$$
$$= \epsilon_2 \qquad (57)$$

$\square$

This corollary reveals that there always exists latent diffusion models that performs at least as well as the autoencoder (which could also be viewed as a traditional regression model). Combining with Theorem B.4, the diffusion models could outperform the traditional regression models.

## C  Experimental Details and Additional Results

### C.1  Architecture design and implementation details

**Graph self-attention with augmented edges.**  As explained in Section 4.2 and Section 4.3, we need a single model to jointly represent and update node, edge and graph features. For denoising network, it should to able to compute scores of all $n^2$ possible edges as we want to generate both the structure and the features of the graph. We therefore design a special graph transformer with augmented edges, which shares a philosophy similar to GRIT [Ma et al., 2023] and EGT [Hussain et al., 2021]. Formally,

$$\boldsymbol{e}_{ij}^{l+1} = \sigma\Big(\rho\big(\kappa(\boldsymbol{Q}\boldsymbol{x}_i^l, \boldsymbol{K}\boldsymbol{x}_j^l) \odot \boldsymbol{E}_w \boldsymbol{e}_{ij}^l\big) + \boldsymbol{E}_b \boldsymbol{e}_{ij}^l\Big) \in \mathbb{R}^{d'}$$
$$\alpha_{ij} = \mathrm{Softmax}_{j \in [n]}(\boldsymbol{W}_A \boldsymbol{e}_{ij}^{l+1}) \in \mathbb{R} \qquad (58)$$
$$\boldsymbol{x}_i^{l+1} = \sum_{j \in [n]} \alpha_{ij}(\boldsymbol{V}\boldsymbol{x}_j^l + \boldsymbol{E}_v \boldsymbol{e}_{ij}^{l+1}) \in \mathbb{R}^{d'}$$

where $l$ represents the number of attention layers, which has been omitted for simplicity in the weight matrices. The matrices $\boldsymbol{Q} \in \mathbb{R}^{d' \times d}$, $\boldsymbol{K} \in \mathbb{R}^{d' \times d}$, $\boldsymbol{V} \in \mathbb{R}^{d' \times d}$, $\boldsymbol{E}_w \in \mathbb{R}^{d' \times d}$, $\boldsymbol{E}_b \in \mathbb{R}^{d' \times d}$,

$\boldsymbol{W}_A \in \mathbb{R}^{1 \times d'}$, $\boldsymbol{E}_v \in \mathbb{R}^{d' \times d}$ are all learnable weight matrices in the attention mechanism. The attention kernel $\kappa$ can be an element-wise addition, as seen in [Ma et al., 2023], or the commonly used Hadamard product. The symbol $\odot$ denotes the Hadamard product (element-wise multiplication). The optional non-linear activation function $\sigma$ can be applied, such as ReLU, or the identity mapping can be used if no non-linear activation is desired. The function $\rho(\boldsymbol{x})$ is optional and defined as $\sqrt{\mathrm{ReLU}(\boldsymbol{x})} - \sqrt{\mathrm{ReLU}(-\boldsymbol{x})}$, which aims to stabilize the training by preventing large magnitudes in the gradients.

To reduce computational complexity, we provide a simplified version where $\boldsymbol{W}_A$ is set to $\boldsymbol{1} \in \mathbb{R}^{d'}$, which suffices to sum the latent dimension. In this case, the attention mechanism reduces to the standard inner product if the kernel $\kappa$ is chosen to be the Hadamard product. Additionally, in simplified attention, $\boldsymbol{E}_v$ can also be set to an identity matrix.

In the edge-enhanced transformer described in Equation (58), we explicitly maintain representations for all nodes and augmented edges. These representations are denoted as $\boldsymbol{X} \in \mathbb{R}^{n \times d}$ and $\boldsymbol{A} \in \mathbb{R}^{n \times n \times d}$, where $\boldsymbol{X}_i = \boldsymbol{x}_i$ and $\boldsymbol{A}_{i,j} = \boldsymbol{e}_{ij}$, with $d$ being the hidden dimension in the corresponding layer. The attention mechanism can be naturally extended to a multi-head case, similar to standard attention. Additionally, residual connections and feedforward layers can be incorporated for both nodes and edges. When representing graph-level attributes, we have two options: (a) utilizing the virtual node trick, where a virtual node is used to represent the graph-level feature; or (b) obtaining the graph-level feature by pooling the node/edge features. It is straightforward to verify that Equation (58) exhibits permutation equivariance for both nodes and edges, and permutation invariance for graph-level attributes.

**Cross attention mechanism for general conditions.** In conditional generation, we need to encode the conditions into the denoising networks. Cross attention is a widely adopted method [Rombach et al., 2021]. We propose a cross-attention mechanism specifically for graphs, which is implemented by

$$\boldsymbol{x}_i^{l+1} = \mathrm{softmax}\left(\frac{(\boldsymbol{Q}_h \boldsymbol{x}_i^l)(\boldsymbol{K}_h \tau(\boldsymbol{y}))^\top}{\sqrt{d'}}\right) \cdot \boldsymbol{V}_h \tau(\boldsymbol{y})$$

$$\boldsymbol{e}_{ij}^{l+1} = \mathrm{softmax}\left(\frac{(\boldsymbol{Q}_e \boldsymbol{e}_{ij}^l)(\boldsymbol{K}_e \tau(\boldsymbol{y}))^\top}{\sqrt{d'}}\right) \cdot \boldsymbol{V}_e \tau(\boldsymbol{y})$$

(59)

where $\boldsymbol{Q}_h, \boldsymbol{Q}_e \in \mathbb{R}^{d' \times d}$, $\boldsymbol{K}_h, \boldsymbol{V}_h, \boldsymbol{K}_e, \boldsymbol{V}_e \in \mathbb{R}^{d' \times d_\tau}$ are learnable weight matrices. Residual connections and feedforward layers are also allowed. The cross-attention mechanism is able to process multiple conditions with different data structures, and becomes an addition in the special case where there is only one condition.

**Cross attention mechanism for graphs.** As stated in Section 4.4, there are cases where the conditions are a graph with part of its features masked. We now propose a novel cross attention mechanism to compute scores of the graph to be generated conditioning on the masked graph.

We assume the latent dimension of the conditioning graph and that of the graph to be generated are identical (otherwise we could use linear layers to project the features of the conditioning graph), and use characters without superscript $c$ to denote the graph being generated from noise (the superscript $l$ is still the number of layer), then

$$\boldsymbol{e}_{ij}^{l+1} = \sigma\left(\rho\left(\kappa(\boldsymbol{Q}\boldsymbol{x}_i^l, \boldsymbol{K}\boldsymbol{x}_j^c) \odot \boldsymbol{E}_w(\boldsymbol{e}_{ij}^l + \boldsymbol{e}_{ij}^c)\right) + \boldsymbol{E}_b \boldsymbol{e}_{ij}^c + \boldsymbol{G}_e \boldsymbol{g}^c\right) \in \mathbb{R}^{d'}$$

$$\alpha_{ij} = \mathrm{Softmax}_{j \in [n]}(\boldsymbol{W}_A \boldsymbol{e}_{ij}^{l+1}) \in \mathbb{R}$$

$$\boldsymbol{x}_i^{l+1} = \sum_{j \in [n]} \alpha_{ij}(\boldsymbol{V}\boldsymbol{x}_j^c + \boldsymbol{E}_v \boldsymbol{e}_{ij}^{l+1}) + \boldsymbol{W}_h \boldsymbol{x}_i^c + \boldsymbol{G}_h \boldsymbol{g}^c \in \mathbb{R}^{d'}$$

(60)

where learnable weight matrices $\boldsymbol{Q}, \boldsymbol{K}, \boldsymbol{V}, \boldsymbol{E}_w, \boldsymbol{E}_b \in \mathbb{R}^{d' \times d}$, $\boldsymbol{W}_A \in \mathbb{R}^{1 \times d'}$, $\boldsymbol{E}_v \in \mathbb{R}^{d' \times d'}$, attention kernel $\kappa$, activation $\sigma$ and signed-square-root operation $\rho$ are identical to those in Equation (58); the new notations $\boldsymbol{G}_e, \boldsymbol{G}_v, \boldsymbol{W}_h \in \mathbb{R}^{d' \times d}$ are also learnable weight matrices. The cross-attention Equation (60) and the self-attention Equation (58) have two major differences: (a) cross-attention uses the features of the conditioning masked graph as the keys and values of the attention; (b) to make the model aware of the node/edge correspondence between the conditioning graph and the main

Table 5: Overview of the datasets used in the paper.

| Dataset | #Graphs | Avg. # nodes | Avg. # edges | Prediction level | Prediction task | Metric |
|---|---|---|---|---|---|---|
| QM9 | 130,000 | 18.0 | 37.3 | graph | regression | Mean Abs. Error |
| ZINC | 12,000 | 23.2 | 24.9 | graph | regression | Mean Abs. Error |
| ogbg-molhiv | 41,127 | 25.5 | 27.5 | graph | binary classif. | AUROC |
| Cora | 1 | 2,708 | 10,556 | edge | binary classif. | accuracy |
| PubMed | 1 | 19,717 | 88,648 | edge | binary classif. | accuracy |
| Cora | 1 | 2,708 | 10,556 | node | 7-way classif. | accuracy |
| PubMed | 1 | 19,717 | 88,648 | node | 3-way classif. | accuracy |
| Physics | 1 | 34,493 | 495,924 | node | 5-way classif. | accuracy |
| Photo | 1 | 7,650 | 238,162 | node | 8-way classif. | accuracy |
| OGBN-Arxiv | 1 | 169,343 | 1,166,243 | node | 40-way classif. | accuracy |

graph to be generated, cross-attention has a node-wise and edge-wise addition of the conditioning node/edge features $x_i^c, e_{ij}^c$ to the node/edge features $x_i^l, e_{ij}^l$ of the main graph.

## C.2 Additional experiments

In this subsection, we provide further experimental results in additional to the ones in the main text. The datasets of these additional experiments include QM9 [Ramakrishnan et al., 2014], small molecular graphs from ogbg-molhiv [Hu et al., 2020], and two citation networks from Planetoid (Cora and PubMed) [Yang et al., 2016]. The statistics of these datasets are summarized in Table 5.

We summarize the results as follows.

- SOTA performance. Our LGD achieves state-of-the-art performance on almost all of these datasets, which verifies that diffusion models can outperform traditional regression and classification models. Our advantages over traditional models are generally more significant when the datasets are large, since diffusion models are capable of modeling complex distributions and empirically perform well with sufficient training data. This implies the strong performance, scalability and generalization of LGD.

- Strong scalability. Our extensive experiments show that LGD could be extremely efficient with proper design, which can completely scale to **extremely large graphs with over 169K nodes such as OGBN-Arxiv**. Moreover, LGD is efficient in both memory and time consumption: all the experiments are carried out on a single RTX 3090 GPU, and both training and inference procedures are fast (e.g. 0.2s/epoch on Cora).

- Flexibility and universality. These experiments with tasks of different levels of types again verifies the flexiblity and universality of our framework. We can always strike a good balance between scalability and performance for all tasks.

**Large-scale generation.** The generation results on large-scale molecule dataset MOSES [Polykovskiy et al., 2020] reported in Table 6 show that LGD has superior performance compared with DiGress in terms of validity and novelty metrics. Apart from DiGress and ConGress, we are the only model to scale to this large dataset, bridging the gap between diffusion-based one-shot graph generation model and other traditional methods like SMILES-based, fragment-based, and autoregressive models.

**Graph-level regression.** We still choose the QM9 dataset for the graph-level regression tasks. Baseline models include MPNN, DTNN [Wu et al., 2017], DeepLRP [Chen et al., 2020], PPGN [Maron et al., 2019], Nested GNN [Zhang and Li, 2021], and 4-IDMPNN [Zhou et al., 2023b]. We consider the same six properties as in the conditional generation task. The results are reported in Table 7. LGD achieves SOTA performance in 5 out of 6 properties, and the advantages are significant. This is an inspiring evidence that generative models could outperform traditional models under our framework.

Table 6: Large-scale generation on MOSES [Polykovskiy et al., 2020] dataset.

| Model | Class | Validity | Uniqueness | Novelty | FCD |
|-------|-------|----------|------------|---------|-----|
| VAE | SMILES | 97.7 | 99.8 | 69.5 | 0.57 |
| JT-VAE | Fragment | 100 | 100 | 99.9 | 1.00 |
| GraphINVENT | Autoreg. | 96.4 | 99.8 | - | 1.22 |
| ConGress | One-shot | 83.4 | 99.9 | 96.4 | 1.48 |
| DiGress | One-shot | 85.7 | 100 | 95.0 | 1.19 |
| LGD (ours) | One-shot | 97.4 | 100 | 95.9 | 1.42 |

Table 7: QM9 regression results (MAE ↓). Highlighted are **first**, **second** best results.

| Target | MPNN | DTNN | DeepLRP | PPGN | Nested GNN | 4-IDMPNN | LGD (ours) |
|--------|------|------|---------|------|------------|----------|------------|
| $\mu$ | 0.358 | 0.244 | 0.364 | **0.231** | 0.433 | 0.398 | **0.088** |
| $\alpha$ | 0.89 | 0.95 | 0.298 | 0.382 | 0.265 | **0.226** | **0.130** |
| $\epsilon_{\text{HOMO}}$ | 0.1472 | 0.1056 | **0.0691** | 0.0751 | 0.0759 | 0.0716 | **0.0363** |
| $\epsilon_{\text{LUMO}}$ | 0.1695 | 0.1393 | 0.0754 | 0.0781 | **0.0751** | 0.0778 | **0.0276** |
| $\Delta\epsilon$ | 0.1796 | 0.3048 | **0.0961** | 0.1105 | 0.1061 | 0.1083 | **0.0395** |
| $c_{\text{v}}$ | 0.42 | 0.27 | 0.129 | 0.184 | **0.0811** | 0.0890 | **0.0817** |

**Graph-level classification.** For graph-level classification task, we choose the ogbg-molhiv [Hu et al., 2020] dataset which contains 41k molecules. The task is a graph binary classification to predict whether a molecule inhibits HIV virus replication or not, and the metric is AUC. We use the official split of the dataset. PNA [Corso et al., 2020], DeepLRP [Chen et al., 2020], NGNN [Zhang and Li, 2021], KP-GIN [Feng et al., 2022a], $I^2$-GNN [Huang et al., 2022], CIN [Bodnar et al., 2021] and SUN(EGO) [Frasca et al., 2022] are selected as baselines. As shown in Table 8, LGD exhibits comparable performance compared with state-of-the-art models. A possible reason is that ogbg-molhiv adopts scaffold split which is not a random split method, leading to imbalance samples in train/validation/test sets. As generative models directly model the distribution, a distribution shift may negatively influence the performance.

Table 8: Ogbg-molhiv results (AUC ↑). Shown is the mean ± std of 5 runs.

| Method | Test AUC |
|--------|----------|
| PNA | $79.05 \pm 1.32$ |
| DeepLRP | $77.19 \pm 1.40$ |
| NGNN | $78.34 \pm 1.86$ |
| KP-GIN+-VN | $78.40 \pm 0.87$ |
| $I^2$-GNN | $78.68 \pm 0.93$ |
| CIN | $\mathbf{80.94} \pm 0.57$ |
| SUN(EGO) | $80.03 \pm 0.55$ |
| GPS | $78.80 \pm 1.01$ |
| LGD-DDPM (ours) | $78.49 \pm 0.96$ |

**Node and edge classification.** The results of node and edge classification on ciatation networks Cora and PubMed are reported in Table 9. The common $60\%, 20\%, 20\%$ random split is adopted. LGD significantly outperforms all the baselines, including OFA [Liu et al., 2024] which uses LLMs as augmented features.

## C.3 Experimental settings

We now provide more details of the experiments in Section 6, including experimental settings and implementation details.

### C.3.1 General settings

**Diffusion process.** In all experiments, we use a diffusion process with $T = 1000$ diffusion steps, parameterized by a linear schedule of $\alpha_t$ and thus a decaying $\bar{\alpha}_t$. For inference, we consider both (a) DDPM; (b) DDIM with 200 steps and $\sigma_t = 0$. Instead of predicting the noises, we use the parameterization where denoising (score) networks $\epsilon_\theta$ is set to directly predict the original data $\mathbf{x}_0$. We set $\lambda = 1$ in Equation (18) as most literature for simplicity.

Table 9: Node-level and edge-level classification tasks (accuracy ↑) on two datasets from Planetoid. Reported are mean ± std over 10 runs with different random seeds. Highlighted are **best** results.

| Dataset Task type | Cora Link | PubMed Link | Cora Node | PubMed Node |
|---|---|---|---|---|
| GCN | $90.40 \pm 0.20$ | $91.10 \pm 0.50$ | $87.78 \pm 0.96$ | $88.9 \pm 0.32$ |
| GAT | $93.70 \pm 0.10$ | $91.20 \pm 0.10$ | $83.00 \pm 0.70$ | $83.28 \pm 0.12$ |
| OFA [Liu et al., 2024] | $94.53 \pm 0.51$ | $98.59 \pm 0.10$ | $74.76 \pm 1.22$ | $78.25 \pm 0.71$ |
| ACM-GCN [Luan et al., 2022] | - | - | $89.13 \pm 1.77$ | $90.66 \pm 0.47$ |
| ACM-GCN+ | - | - | $89.75 \pm 1.16$ | $90.46 \pm 0.69$ |
| ACM-Snowball-3 | - | - | $89.59 \pm 1.58$ | $91.44 \pm 0.59$ |
| LGD (ours) | $96.48 \pm 0.17$ | $99.03 \pm 0.08$ | $93.91 \pm 0.55$ | $92.88 \pm 0.29$ |

**Model architectures.** In our main paper, we introduce the formulation where we generate the full $\mathbf{A} \in \mathbb{R}^{n \times n \times d}$, so that we have a unified framework that can tackle tasks of all levels. However, this does not necessarily mean that we always need to generate $\mathbf{A}$—instead, we can generate only the features that we desire. Correspondingly, although we use the specially designed graph transformers and cross-attention mechanism for denoising networks, other architectures (e.g. general GNNs) are also applicable, as we have already mentioned in the main text (see Section 4.1). For example, in node classification tasks where we only need to predict the labels of nodes, we can generate only the node features, where a simple MPNN (e.g., GCN) would be enough for the denoising network. We implement this setting in some of our experiments, showing that our LGD can **scale to extremely large graphs** with over 169K nodes (e.g., OGBN-Arxiv). The actual training and inference procedures are also very efficient, for example each epoch takes only ~0.2s in Cora dataset.

For all graph-level tasks, we use the encoder $\mathcal{E}_\phi$ with the augmented-edge enhanced graph transformer as described in Equation (58). For all node-level tasks, we use an MPNN model (e.g. GINE [Hu et al., 2019], GCN [Kipf and Welling, 2016] or GAT [Velickovic et al., 2017]) as the encoder. For regression and classification tasks, as the input graph structures are complete, we can also add positional encoding and MPNN modules to the encoder. For all tasks, each task-specific decoder $\mathcal{D}_\xi$ is a single linear layer for reconstruction or prediction. Due to the discussion in Appendix B, we choose the dimension of the latent space from $[4, 8, 16, 32]$. We pretrain the autoencoders separately for each tasks. Pretraining across datasets (especially across domains) is a more challenging setting, but may also benefit the downstream tasks, which would be an interesting future direction.

For denoising (score) network $\epsilon_\theta$, we consider two cases: (i) we want to generate structures and features simultaneously; (ii) we only want to generate3 features. In the first case, due to the reason explained in Section 4.3, MPNNs and PEs are ill-defined in the corrupted latent space $\mathcal{H}_t$ and are hence not applicable. Therefore, we use pure graph transformer architectures for $\epsilon_\theta$. We classify all generation tasks and graph-level prediction tasks into this case. For unconditional generation, each layer computes self-attention as in Equation (58). For conditional generation, each module consists of a self-attention layer, a cross-attention layer, and another self-attention layer. For QM9 conditional generation tasks where the conditions are molecular properties, we use an MLP as the conditioning encoder $\tau$ and the cross-attention in Equation (59). For regression on Zinc and classification on ogbg-molhiv, since the conditions are masked graphs, we use the specially designed cross-attention in Equation (60). In these cases, the conditioning encoder $\tau$ is shared with the graph encoder $\mathcal{E}_\phi$. To enable good representation of both masked graphs and the joint representation of graphs with labels, we randomly mask the labels in the pretraining stage, so that both $\mathcal{E}_\phi(\boldsymbol{X}, \boldsymbol{A})$ and $\mathcal{E}_\phi(\boldsymbol{X}, \boldsymbol{A}, \boldsymbol{y})$ have small reconstruction errors - though the latter is typically much smaller than the former one. We do not add virtual nodes in QM9 datasets, and the graph feature is obtained by (sum or mean) pooling the nodes (and optionally, edges). For Zinc and ogbg-molhiv, we add a virtual node where we embed the graph-level label via the encoder to obtain target refined full graph representations, and we use an embedding layer for class labels and an MLP for regression labels. All networks have ReLU activations, and we use layer norm instead of batch norm for all transformers for training stability. However, in case (ii), as the structures are clearly defined and known, we can still utilize the MPNN

Table 10: Ablation study on latent dimension on Zinc (MAE ↓). Shown is the mean ± std of 3 runs.

| Latent dimension | Test MAE |
|---|---|
| 4 | $0.081 \pm 0.006$ |
| 8 | $0.080 \pm 0.006$ |
| 16 | $0.084 \pm 0.007$ |

family denoising models. We classify all node-level tasks into this case so that we could use MPNNs with $O(n)$ complexity to avoid the extensive computation of attention mechanisms.

### C.3.2 Task specific settings

**Unconditional generation on QM9.** Here we provide other hyperparemeters for the experiments of QM9 unconditional generation. We use an encoder with 96 hidden features and 5 layers. The denoising network has 256 hidden channels and 6 self-attention layers. We train the model for 1000 epochs with a batch size of 256. We adopt the cosine learning rate schedule with 50 warmup epochs, and the initial learning rate is $1e - 4$.

**Conditional generation on QM9.** We train a separate model for each target. The encoders have 3 layers and 96 hidden channels. The denoising networks have 4 cross-attention modules (8 self-attention layers and 4 cross-attention layers in total) and 128 hidden channels. We train for 1000 epochs with a batch size of 256, and the cosine learning rate schedule with initial learning rate $1e - 4$.

**Regression on Zinc.** The encoder has 64 hidden channels and 10 layers, augmented with GINE [Hu et al., 2019] and RRWP positional encodings [Ma et al., 2023]. The score network has 64 hidden channels and 4 cross-attention modules (8 self-attention layers and 4 cross-attention layers in total). We train the diffusion model for 2000 epochs with a batch size of 256, and a cosine learning rate with 50 warm-up epochs and initial value $1e - 4$. We also conduct ablation studies on Zinc, see Appendix C.4.

**Classification on ogbg-molhiv.** Same as Zinc, the encoder has 64 hidden channels and 10 layers, augmented with GINE [Hu et al., 2019] and RRWP positional encodings [Ma et al., 2023]. The score network also has 64 hidden channels and 4 cross-attention modules. We train the diffusion model for 500 epochs with a batch size of 256, and a cosine learning rate with 20 warm-up epochs and initial value $1e - 4$. We observe overfitting phenomenon in the early stage, which may be caused by the imbalanced distribution between training set and test set, see Appendix C.5 for discussions.

**Classification on node-level datasets.** We use the combination of GCN [Kipf and Welling, 2016] and GAT [Velickovic et al., 2017] as the encoder, with 7 layers and 160 hidden dimensions. We use LapPE and RWSE. The score network has 192 hidden channels and 4 cross-attention modules. We train the diffusion model for 500 epochs with a batch size of 1, and a cosine learning rate with 50 warm-up epochs and initial value $1e - 4$.

### C.4 Ablation studies

We conduct ablation studies on Zinc dataset to investigate (a) the impacts of the dimension $d$ of the latent embedding; (b) whether ensemble technique helps to improve the prediction quality as the generation is not deterministic.

**Dimension of the latent embedding.** We use encoders with output dimension (i.e. the dimension of the latent embedding for LGD) $4, 8, 16$ respectively, and inference with the fast DDIM for 3 runs with different random seeds in each setting.

According to the results in Table 10, LGD-DDIM reveals similar performance with these latent dimensions, while $d = 16$ leads to a slight performance drop. This coordinates with our theoretical analysis in Appendix B, as large dimensions of latent space may cause large diffusion errors. It would

Table 11: Ablation study of ensembling on Zinc (MAE ↓). Shown is the mean ± std of 3 runs.

| # of ensembled predictions | Test MAE |
|---|---|
| 1 | $0.081 \pm 0.006$ |
| 5 | $0.081 \pm 0.006$ |
| 9 | $0.079 \pm 0.006$ |

be interesting to see the performances in a much smaller (e.g. $d = 1$) or much larger (e.g. $d = 64$) latent space.

**Ensemble of multiple generations.** One possible issue with solving regression and classification tasks with a generative model is that the output is not deterministic. To investigate whether the randomness improve or hurt the performance, we test with DDIM using a same trained model which ensembles over $1, 5, 9$ predictions respectively. We use the median of the multiple predictions in each case.

As shown in Table 11, ensembling over multiple predictions has almost no impacts on the performance. This is somewhat surprising at the first glance. It shows that the diffusion model can often output predictions around the mode with small variances, thus modeling the distributions well. As mentioned in the main text, the SDE-based DDPM also outperforms ODE-based DDIM, aligns with previous empirical and theoretical studies that the randomness in diffusion models could lead to better generation quality [Cao et al., 2023].

### C.5 Experimental findings and discussions

**Evaluation metrics in unconditional molecule generation.** We now further discuss the evaluation metrics of unconditional molecule generation with QM9 dataset. In the main text we report the widely adopted validity and uniqueness metrics as most literature do. However, it is notable that the validity reported in Table 1 (and also in most papers) is computed by building a molecule with RdKit and calculating the proportion of valid SMILES string out of it. As explained in Jo et al. [2022], QM9 contains some charged molecules which would be considered as invalid by this method. They thus compute a more relaxed validity which allows for partial charges. As a reference, GDSS-seq and GDSS Jo et al. [2022] achieve $94.5\%$ and $95.7\%$ relaxed validity, respectively; in comparson, our LGD achieves $95.8\%$ relaxed validity, which is also better. We do not report the novelty result for the same reasons as explained in [Vignac et al., 2022]. This is because QM9 is an exhaustive enumeration of the small molecules that satisfy certain chemical constrains, thus generating novelty molecules outside the dataset may implies the model has not correctly learned the data distribution.

**Experimental findings.** In our experiments, we observe many interesting phenomenon, which are listed below. We hope these findings could give some insights into further studies.

- Regularization of the latent space. We experimentally find that exerting too strong regularization to the latent space, e.g. a large penalty of KL divergence with the standard Gaussian, would lead to performance drop of the diffusion model. This is understandable as the latent space should not only be simple, but also be rich in information. Strong regularizations will cause the model to output a latent space that is too compact, thus hurt the expressiveness. We find in our experiments that a layer normalization of the output latent representations tends to result in better generation qualities, while KL or VQ regularization are not necessary.

- Rapid convergence. We find that our latent diffusion model tends to converge rapidly in the early training stage. Compared with non-generative models such as GraphGPS [Rampásek et al., 2022], LGD typically needs only $\frac{1}{5} \sim \frac{1}{3}$ number of epochs of them to achieve similar training loss, which makes the training of LGD much faster. We attribute this to the advantages of the powerful latent space, as the diffusion model only needs to "refine" the representation on top of the condition that is already close to ground truth.

- Generalization. Interestingly, we find that while LGD usually have good predictions (in both train and test set) in the early stage of training, the test loss may increase in the final stages, forming a "U" curve in test loss. This is obviously a sign of overfitting due to the

strong capacity of diffusion models to model complex distributions. If the distribution shift between train and test set becomes larger (e.g. ogbg-molhiv compared with Zinc), the overfitting phenomenon gets more obvious. This implies that LGD is extremely good at capturing complex distributions, and thus has the potential to scale well to larger datasets that fits with the model capacity.

# D    Discussions

## D.1    More discussions on graph generative models

Among graph generative models, although some auto-regressive models show better performance, they are usually computationally expensive. Moreover, one worse property of auto-regressive graph generative models cannot maintain the internal permutation invariance of graph data. On the other hand, some one-shot graph generative models are permutation invariant, but those based on VAEs, normalizing flows and GANs tend to under-perform autoregressive models. Currently, graph diffusion generative models are the most effective methods, but they have to overcome the difficulty brought by discreteness of graph structures and possibly attributes. Consequently, [Niu et al., 2020] and [Jo et al., 2022] can only generate the existence of edges via comparing with a threshold on the continuous adjacency matrix $\mathbf{A} \in \mathbb{R}^{n \times n}$. Besides, [Niu et al., 2020] cannot deal with node or edge features, while [Jo et al., 2022] can only handle node attributes. DiGress [Vignac et al., 2022] operates on discrete space, which may have positive impacts on performance in some cases where discrete structures are important (e.g. molecule generation or synthetic graph structure data). However, DiGress is only suitable for discrete and categorical data, while they fail to handle continuous features. All these score-based or diffusion models are unable to provide all types of node and edge (and even graph) features, including categorical and continuous ones. In comparison, our model is able to handle features of all types including continuous ones. Moreover, our continuous formulation is empirically more suitable for well established classical diffusion models compared with discrete space. Our method also differs from Xu et al. [2023], where their model can only generate nodes and requires additional computations to predict edges based on the generated nodes. Overall, our LGD is the first to enable generating node, edge and graph-level features of all categories in one shot.

Some concurrent works provide some insights and implications on graph generation. Yang et al. [2023b] incorporates data-dependent, anisotropic and directional noises in the forward diffusion process to alleviate transforming anisotropic signals to noise too quickly. The denoising network directly predicts the input node features from perturbed graphs, and Yang et al. [2023b] also propose to use diffusion models for unsupervised graph representation learning by extracting the feature maps obtained by the decoder. Dan et al. [2023] estalishes a connection between discrete GNN models and continuous graph diffusion functionals through Euler-Lagrange equation. On top of this, Dan et al. [2023] propose two new GNN architectures: (1) use total variation and a new selective mechanism to reduce oversmoothing; (2) train a GAN to predict the spreading flows in the graph through a neural transport equation, which is used to alleviate the potential vanishing flows.

## D.2    Building the pretrain, finetune and in-context learning framework upon LGD

**A complete training framework.**    We now describe the overall architecture and training procedure of the general purposed graph generative model that can be built upon LGD, and discuss the potential of LGD as the framework for future graph foundation models. As described before, the entire model consists of a powerful graph encoder $\mathcal{E}$ and latent diffusion model $\theta$, which can be shared across different tasks and need to be pretrained - although we currently train them separately for each task. As for different downstream tasks, we can adopt finetuning or prompt tuning strategies. In both cases, we need to train a lightweight task-specific graph decoder $\mathcal{D}$. We need to update the parameters of $\mathcal{E}$ and $\theta$ while finetuning. In prompt tuning, however, we keep these pretrained parameters fixed and only train a learnable prompt vector concatenated with input features of prompt graphs, which enables in-context learning ability for pretrained graph models. Notably, some prior work like PRODIGY [Huang et al., 2023] conduct in-context learning by measuring the "similarity" between test and context examples, which is different from the conditional generation and prompting as in NLP and LGD.

A promising future direction is to pretrain the graph encoder $\mathcal{E}$ and the generative model $\theta$ across a wide range of tasks. These two parts can be trained simultaneous or sequential, as well as through

different paradigm. Here, $\mathcal{E}$ includes the initial tokenization layer as well as a powerful graph learner backbone (e.g. a GNN or graph transformer) that encodes input graph into a semantic meaningful latent space for all downstream tasks. Pretrain of $\mathcal{E}$ consists of both unsupervised learning and supervised learning trained on downstream tasks with labels. The unsupervised loss includes both contrastive loss $\mathcal{L}_{contrast}$ and reconstruction loss of self-masking $\mathcal{L}_{recon}$. The supervised learning is jointly trained with a downstream classifier or regression layer (that will not be used later), according to classification entropy loss $\mathcal{L}_{ce}$ or regression MSE/MAE loss $\mathcal{L}_{regress}$.

We train the diffusion model $\theta$ in a supervised manner. Extra tokens are applied to represent masked values or attributes that need to be predicted. In traditional generation tasks, the training procedure is similar to stable-diffusion, where we compute loss between predicted denoised graph and the input graph in the latent space encoded by $\mathcal{E}$. As for classification and regression tasks, we only input part of graph features, and expect the model to reconstruct the input features while predicting those missing features. For example, in a link prediction task, the input graph contains all node features and only part of edges. Then the model need to reconstruct the known input features and simultaneously predict the rest links. Therefore, the loss is the summation of reconstruction loss (in the latent space) and the corresponding classification or regression loss.

For tasks of different levels and types, note that the reconstructed graph representation in the latent space contains all node and edge features that are powerful enough for any downstream tasks. Thus only a (lightweight) task specific decoder $\mathcal{D}$ is needed, like a classifier for a classification task and a regression layer for a regression task. The node and edge level attributes can be directly obtain based on corresponding latent representation, while graph attributes is available by pooling operations, or by adding a virtual node with trainable graph token for every task. To train a task specific decoder, we can either finetune along with the entire pretrained model on the target dataset (whole dataset and few shot are both possible), or only train them with prompt vectors while fixing the parameters of pretrained graph encoder $\mathcal{E}$ and generative model $\theta$.

**Domain adaptation.** We now present the potential of our framework in domain adaptation, while leaving the experiments for future work. To deal with graphs from various domains, we create a graph dictionary for all common node and edge types in graph datasets, such as atom name in molecular graphs. The vectorization is via the first embedding layer of the graph encoder network $\mathcal{E}$, which is analogous to text tokenization in LLM. For those entities rich in semantic information, we can also make use of pretrained foundation models from other domains or modals through cross attention mechanism in the latent diffusion model.

After finishing pretraining the powerful graph encoder $\mathcal{E}$ and the latent generative model $\theta$, we need to adapt the entire model to target tasks. Previously, a widely adopted paradigm in graph learning field is the pretrain-finetune framework. During the finetune procedure, we jointly train the task specific decoder $\mathcal{D}$, and finetune $\mathcal{E}$ and $\theta$ on the target task. However, finetuning has at least the following drawbacks: (1) significantly computation cost compared with prompting, since we need to compute gradients of all parameters; (2) may lead to catastrophic forgetting of the pretrained knowledge, thus performing worse on other tasks and possibly requiring one independent copy of model parameters for each task.

Prompting is another strategy to adapt the pretrained model to downstream tasks, which designs or learns a prompt vector as input while keeping the pretrained $\mathcal{E}$ and $\theta$ fixed. Compared with finetuning, prompting requires much less computation resources since we only need to learn an input vector. Here we introduce a graph prompt mechanism which is the first to allow for real in-context learning. In the target task, we provide prompt example graphs with labels, whose input node and edge features are concatenated with a learnable task-specific prompt vector. The prompt vectors are learned through this few shot in-context learning, which adapt the model better to the task without retraining or finetuning. Besides, to keep the hidden dimension aligned, we also concatenate a prompt vector in the pretrain and finetune stages, which can be serve as initialization of downstream prompt vectors in similar tasks.

### D.3 Complexity of LGD and comparison with traditional models

As is shown in Appendix C empirically, LGD has good scalability with proper settings. We now provide a thorough discussion on the computation complexity and scalability of LGD.

The complexity of diffusion models mainly depends on: (i) the internal complexity of the denoising networks; (ii) the complexity brought by the diffusion process. For (i) the complexity of the denoising networks, our LGD framework is compatible with flexible network architectures, thus could always strike a good balance between complexity and expressivity. We consider both the generation tasks and prediction tasks. In the generation setting, we have $n^2$ complexity as we utilize our proposed graph transformers, which is generally unavoidable while generating from scratch. It is remarkable that previous models also have large complexity. Actually, GDSS [Jo et al., 2022] also models the entire $n \times n$ adjacency matrix; GeoLDM [Xu et al., 2023] and EDM [Hoogeboom et al., 2022] both use EGNN, resulting to $n^2$ computation complexity as well; Digress [Vignac et al., 2022] also uses a graph transformer with $n^2$ complexity. However, in the prediction settings (e.g. graph-level and node-level classification and regression tasks), we could leverage arbitrary GNN models when the graph structures are clearly defined - we only need to generate features. The complexity of LGD thus reduces to $O(n)$ if we use MPNN models, which enables LGD to scale to extremely large graphs just as traditional graph models do.

For (ii) the complexity of the diffusion process, since in the training stage we always sample only one diffusion step, the complexity of diffusion models do not differ from traditional models (need only one forward pass of the denoising networks). In the inference stage, while the diffusion model needs to iteratively denoise, the inference mode is internally more efficient than training mode. We have also shown that LGD is capable of incorporating efficient inference methods including DDIM, which requires much less inference steps compared with $T$. Moreover, for easier tasks we can set $T$ a relatively small value, which also helps to improve the overall efficiency.

Superior training efficiency of LGD reported in Appendix C is also remarkable. Actually as we discussed in Appendix C.5, we empirically observe that **the diffusion models have better training efficiency compared with traditional regression or classification models**. This may due to the fact that compared with traditional models, diffusion models can learn the distribution in a very efficient way by through iterative sampling, which decompose the complex distribution transformation into easier steps. LGD also benefit from the powerful encoder, which uses the latent embedding containing label information as the training target of diffusion models. In detail, the diffusion models tries to recover $\mathcal{E}(x, y)$ conditioning on $\mathcal{E}(x)$. When the encoder already have good representations (i.e. $\mathcal{E}(x)$ is a good estimator of $y$), the training of diffusion model can be extremely efficient. As our theory in Appendix B reveals, **the latent diffusion models can outperform the encoders** (which can be regarded as traditional regression or classification models) with some mild assumptions hold. Besides the rapid convergence in early training stage, we also observe some overfitting phenomenon (especially on smaller datasets), which indicate LGD's power in learning distributions and its potential to perform better when scale to larger datasets - it would be an interesting future direction to observe whether there would be emergence ability of diffusion-based generative models like LGD.

Moreover, we can always sample subgraphs for the large graphs so that even the full graph transformers would be applicable. In summary, our framework is general and extremely flexible. We can always balance between the complexity and effectiveness according to different task settings.

### D.4 Discussion on solving prediction tasks with diffusion models

Although Li et al. [2023a] also mention the idea of doing classification with diffusion models, their formulation is complex and computational costly as they have to enumerate all classes. In our framework, the diffusion model directly learns the conditional distribution $p(y|x)$, while the diffusion models in [Li et al., 2023a] still learns $p(x|y)$ and they rely on Bayes' theorem to inference $p(y|x)$. Therefore, our model is more efficient since we only need one forward pass of diffusion models, while [Li et al., 2023a] needs $N$ passes where $N$ is the number of classes (which could be very large). Moreover, their setting is only suitable for classification, as they cannot apply Bayes' theorem to continuous prediction targets, thus failing to handle regression tasks. In comparison, our formulation is much simpler and more efficient, and is applicable to both regression and classification tasks. We also derive error bounds for the conditional latent diffusion models in these tasks, see Appendix B.

It is also interesting to explore the possibility of applying diffusion models to regression and classification tasks in other fields, such as computer vision. It remains to see whether diffusion models could outperform traditional non-generative models in all task types, both inside and outside the graph learning field.

