# OpenReview forum: "Unifying Generation and Prediction on Graphs with Latent Graph Diffusion"
_NeurIPS.cc/2024/Conference — NeurIPS 2024 poster_

### Official Review · Reviewer_PVEb · 2024-07-08

**Soundness:** 2
**Presentation:** 3
**Contribution:** 2
**Rating:** 5
**Confidence:** 4

**Summary:**

This paper proposes a Latent Graph Diffusion model that unifies multiple tasks such as classification, regression, and generation in a generative task. Additionally, this model achieves feature generation in multiple categories, including node-level, edge-level, and graph-level.

**Strengths:**

1. This paper attempts to address a crucial task in the field of graphs, which is to propose a model capable of handling various tasks such as classification, regression, and generation at different levels, including node-level, edge-level, and graph-level.

2. The writing in this paper is relatively clear.

**Weaknesses:**

1. The evaluations conducted by the authors are insufficient. For example, in the generation task, they avoid comparing their proposed method with classical diffusion-based graph generation models such as DiGress [1], HGGT [2], and DruM [3], which perform better in terms of metrics like Validity on QM9. Additionally, they avoid comparisons on larger datasets like MOSES [4] or GuacaMol [5]. Furthermore, they do not compare with commonly used metrics for graph generation tasks, such as FCD and NSPDK.

2. Regarding the approach of treating classification and regression tasks as generation tasks, the authors should provide a simple baseline (which can be seen as an ablation study) to demonstrate that they have not unnecessarily complicated simple and intuitive tasks. For example, they could use condition features as input and train a baseline prediction model with the same architecture as the denoising model, using the corresponding masked global features as supervised signals. This would help assess the benefits brought by the generation model.

3. The authors aim to develop a model that can address graph learning tasks at all levels (node, edge, and graph) and types (generation, regression, and classification), which is hard to achieve. The authors are suggested to provide a clear table to include detailed information about the source and size of the training dataset, as well as the number of model architecture and parameters for reference.

4. The explanation of how the decoder of generation tasks reconstructs the structural information, i.e., A, of the graph is not clear.

5. Figure 1, mentioned in the Section 5, is displayed in the Appendix.

[1] C. Vignac, I. Krawczuk, A. Siraudin, B. Wang, V. Cevher, and P. Frossard. DiGress: Discrete denoising diffusion for graph generation. In International Conference on Learning Representations, 2022.

[2] Y. Jang, D. Kim, and S. Ahn. Graph generation with K2-trees. In The Twelfth International Conference on Learning Representations, 2023.

[3] Jaehyeong Jo, Dongki Kim, and Sung Ju Hwang. Graph generation with diffusion mixture., ICML 2024.

[4] D. Polykovskiy, A. Zhebrak, B. Sanchez-Lengeling, S. Golovanov, O. Tatanov, S. Belyaev, R. Kurbanov, A. Artamonov, V. Aladinskiy, M. Veselov, et al. Molecular sets (moses): a benchmarking platform for molecular generation models. Frontiers in pharmacology, 11:565644, 2020.

[5] N. Brown, M. Fiscato, M. H. Segler, and A. C. Vaucher. Guacamol: benchmarking models for de novo molecular design. Journal of chemical information and modeling, 59(3):1096–1108, 2019.

**Questions:**

1.	Whether the Encoder and Decoder are permutation equivariant?
2.	Will the authors release the source code and checkpoints?

**Limitations:**

Yes.

---

> ### Author Rebuttal · Authors · 2024-08-07
>
> We thank the reviewer for the useful comments. We now address your concerns as follows.
>
> 1. Experiments (weakness 1).
>
> * Originally, LGD is designed to be a single framework that can do both generation and regression/classification, where we put our experimental focus on demonstrating that latent diffusion models can achieve state-of-the-art regression/classification results. Now, we have rerun our generation experiments on QM9 and (i) compare our results with state-of-the-art diffusion-based graph generation methods including DiGress [1], HGGT [2] and GruM [3]; (ii) use metrics including FCD and NSPDK to better evaluate the quality of generation. In our new experiments, we train a more powerful encoder/decoder by incorporating positional encodings and more complex reconstruction tasks to obtain better latent graph representations, and carefully choose the hyper-parameters of the diffusion model. The results are shown in Table 1 of the pdf in the global response. It is remarkable that our new results are highly competitive or superior than the baselines in nearly all metrics, which reveals the strong generation ability of LGD. Especially, LGD achieves a better FCD score (in favor of generating graphs similar to the training data) than the three strong recent baselines [1,2,3] while sacrificing the least novelty.
>
> * We are conducting new experiments on the larger dataset MOSES. Since the time is limited, the results are not yet ready. We will update the results as soon as possible, using comments and the link in the global response. We promise to include the new results in our camera-ready paper. It is also notable that most recent studies including HGGT [2] and GruM [3] also do not include these datasets at scale.
>
> 2. Clarifications on treating classification and regression tasks as generation tasks (weakness 2). Actually, in Appendix B of our paper, we already provided solid theoretical analysis and clarifications on why diffusion models can perform better than traditional deterministic models in classification and regression tasks. In detail, Theorem B.4. provides error bounds of diffusion models on regression tasks, equation (52) explicitly states the condition where diffusion models outperform regression models, and Corollary B.6. provides guarantees that there exists at least one latent diffusion model whose MAE is not larger than the autoencoder. Combining all these pieces, we can use arbitrary existing prediction models as the autoencoder, and the diffusion model trained on top this autoencoder is able to perform better than this prediction model. To empirically verify this, we utilize the ZINC dataset and train a regression model with our proposed graph attention architecture (the same as the denoising model) using the masked global features (regression target) as the supervising signal. This prediction model has $0.084\pm 0.004$ test MAE on ZINC, while the diffusion model trained upon the autoencoder with the same architecture achieves $0.065\pm0.003$ MAE. This results verifies that we are not complicating simple tasks---instead, diffusion models can provably outperform traditional prediction models.
>
> 3. Clarification on training (weakness 3). First of all, LGD is a unified framework (and not necessarily a single model) for all levels of graph learning tasks. As explained in both the limitation section and the experiment section of our paper, we currently train our models separately for each task. However, we are indeed able to train a single model for all tasks using LGD. For example, we can use OFA [4] as the autoencoder, which is a single model that embeds graphs from various domains into a unified latent space through a powerful LLM-GNN mixture architecture. We can then train a diffusion model in this latent space to tackle tasks of all levels and all types, utilizing the unified formulation in our paper. We believe the unified formulation of our paper is a significant contribution and a necessary stage towards graph foundation models. As for the model architectures, details are provided in Appendix D.1 and D.3.1. As a reference, the total number of parameters of LGD (including both the autoencoders and the diffusion model) is 1,825,481 for the regression task on ZINC, and 6,605,545 for the generation task on QM9.
>
> 4. Decoding graph structure (weakness 4). As clearly explained in our main text, the latent space contains both the structural information and edge features, since the encoder treats absence of an edge as a special feature. We have a latent representation $W\in \mathbb R^{n\times n\times d}$ for adjacency matrix $A$ (line 145), which is exactly the diffusion target. After obtaining the generated latent edge-augmented representations $\hat W\in \mathbb R^{n\times n\times d}$, we directly apply the pretrained decoder (which is actually a linear layer) to reconstruct both the graph structure ($A$) and edge features. In summary, LGD is a one-shot generation method that generates graph structures and features simultaneously.
>
> 5. Figure (weakness 5). Figure 1 is presented in the Appendix due to limited space. We will move the figure to the main context in the camera ready version.
>
> 6. Equivariance (question 1). The encoder and decoder are indeed permutation equivariant. We can either use existing models such as MPNN and graph Transformers, or use our specially designed graph attention mechanisms as encoder/decoder. All of the above models are permutation equivariant. Actually, we also use MPNN and graph transformers for the denoising network, making our diffusion model also permutation equivariant (in terms of each denoising step).
>
> 7. Source code (question 2). We provide our code through the anonymous link in the global response. We will definitely release all the source code and checkpoints in the final version.

---

> ### Author Response · Authors · 2024-08-07
>
> References
>
> [1] C. Vignac, I. Krawczuk, A. Siraudin, B. Wang, V. Cevher, and P. Frossard. DiGress: Discrete denoising diffusion for graph generation. In International Conference on Learning Representations, 2022.
>
> [2] Y. Jang, D. Kim, and S. Ahn. Graph generation with K2-trees. In The Twelfth International Conference on Learning Representations, 2023.
>
> [3] Jaehyeong Jo, Dongki Kim, and Sung Ju Hwang. Graph generation with diffusion mixture., ICML 2024.
>
> [4] Liu, H., Feng, J., Kong, L., Liang, N., Tao, D., Chen, Y., and Zhang, M. One for all: Towards training
> one graph model for all classification tasks. In The Twelfth International Conference on Learning
> Representations, 2024

---

> ### Author Response · Authors · 2024-08-13
> **We are looking forward to your reply**
>
> Dear reviewer,
> While we have addressed all your concerns and answered all your questions, we are looking forward to your reply. As shown in the global response, we rerun the experiments on QM9 and introduce FCD and NSPDK metrics, showing that LGD is competitive with all the strong baselines you mentioned in terms of all metrics. We also conduct experiments on large scale dataset MOSES and show that LGD outperforms DiGress. The answers to your other questions are included as well in our response. While other reviewers have replied positively, we are looking forward to your reply and we are happy to answer any further question you may have. Thank you.

---

> > ### Comment · Reviewer_PVEb · 2024-08-13
> > **thank you for your rebuttal**
> >
> > Dear authors, thank you for your response. After reviewing the additional experimental results, I decided to increase my score to 5.

---

> > > ### Author Response · Authors · 2024-08-13
> > > **Thank you for your reply**
> > >
> > > Dear reviewer,
> > > we sincerely appreciate your constructive comments and kind reply. We believe that this paper has great contribution to the graph learning and the machine learning community.

---

### Official Review · Reviewer_a5WK · 2024-07-11

**Soundness:** 3
**Presentation:** 2
**Contribution:** 2
**Rating:** 6
**Confidence:** 4

**Summary:**

The paper proposes Latent Graph Diffusion to generate node, edge, and graph-level features to meet the need of different tasks under the unified framework. Extensive experiments show the competitive performance across various graph-based tasks.

**Strengths:**

1. It is good to reformulate regression and classification problems with diffusion generation task.
2. The paper is technically sound, and the introduction of diffusion in latent space is well-motivated.
3. The unified framework has theoretical support.

**Weaknesses:**

1. Readers have to jump to Appendix over and over again to have a general understanding of your proposed model.
2. Despite the unified formulation as generation, other parts of the paper lack novelty. For example, there is a lack of new breakthroughs in specific model design and diffusion models. The underlying diffusion and denoising models largely build upon existing methods.
3. No code or demo available.

**Questions:**

1. How does the LGD framework scale with very large graphs?
2. In the task of node classification from Planetoid, is the LGD model capable of generating a new graph almost the same as Cora or PubMed?
3. The graph will be too dense when considering no link as a special type of edge? How to deal with this problem

**Limitations:**

1. The lack of detailed analysis of time and space complexity makes the performance of the model in practical applications less transparent.
2. The sensitivity analysis of parameters in the conditional generation task is insufficient, especially since the influence of important hyperparameters is not studied deeply.

---

> ### Author Rebuttal · Authors · 2024-08-07
>
> We thank the reviewer for the comments and acknowledgement.
>
> 1. Writing (weakness 1). We will refine the writing of the paper and provide as much important information as possible in the main text.
>
> 2. Novelty (weakness 2). We believe that our unified formulation is novel and beneficial. Regarding model design, we design novel self-attention and cross-attention mechanisms for graphs with augmented edges, as shown in Appendix D.1. For the diffusion model, we explained in our main text that we use the latent diffusion to tackle the problems brought by the discrete structure, feature misalignment and task variety. It is novel to use a latent diffusion model to perform graph learning tasks of all levels and all types. Furthermore, while architecture and diffusion model design are not our main focus, our LGD framework is extremely flexible, being able to incorporate various model designs and diffusion processes (e.g. DDPM, DDIM, Rectified flow etc.).
>
> 3. Source code (weakness 3). We provide the code through the anonymous link in the global response. We will definitely release all the code and checkpoints in the camera-ready version.
>
> 4. Scalability of LGD (question 1 and 3). We have explained the scalabiltiy of LGD in multiple places in appendix, such as architecture design in Appendix D.3, theoretical complexity in Appendix E.2, and experimental verifications in Appendix D.2 (line 887-891). We do not have to always model the dense graph where no link is a special type of edge---actually when we do not need to generate structures, we can utilize the sparse graph and only consider the real edges. We can thus use efficient MPNN based models to scale LGD to extremely large graphs such as OGBN-Arxiv, see Table 3 in the paper.
>
> 5. Node classification (question 2). In classification and regression tasks, generating the entire graphs is not our main focus---LGD mainly aims to generate the class labels of nodes. However, since LGD actually refines the representation in the latent space, it is still able generate the whole graph. LGD outputs the refined representation, which contains the information of both input (graph structures and known features) and class labels to be predicted, thus it could generate new graphs similar to the training sets.

---

> > ### Comment · Reviewer_a5WK · 2024-08-12
> >
> > Thanks to the author for addressing my questions. I have to say that generating the refined representation is one thing, and generating new graphs (graph structure and node features) is one another.  Anyway, I will maintain my positive score.

---

> > > ### Author Response · Authors · 2024-08-12
> > > **Thank you for your reply**
> > >
> > > We sincerely thank the reviewer for the positive reply. Regarding the question, it is true that generating refined representation and generating new graphs are different things. However, since the target of node classification is not generating new graphs, the models are trained in a conditional generation manner, which is different from unconditional sampling for generating new graphs. We believe the capability of LGD to generate new graphs has been verified in unconditional generation experiments presented in the original submission and our rebuttal. Please inform us if you have any questions.

---

### Official Review · Reviewer_XBTL · 2024-07-12

**Soundness:** 3
**Presentation:** 3
**Contribution:** 3
**Rating:** 6
**Confidence:** 4

**Summary:**

This paper proposes a model framework, LGD, which addresses multiple types of graph tasks and can simultaneously handle both generative and predictive tasks. Specifically, the LGD framework employs a method similar to stable diffusion, using a pretrained graph encoder to obtain latent representations and then training a diffusion model in the latent space. To use the diffusion model for predictive tasks, the authors incorporate the label in regression and classification tasks as a partially masked feature, using the rest of the training set as a condition to predict the masked label. Finally, the authors validate their approach through experiments for prediction, classification, and regression tasks.

**Strengths:**

- The idea of using a conditional diffusion model for predictive tasks is novel. Unlike traditional representation learning approaches, LGD can directly generate labels using the diffusion model, providing a new perspective for the community.

- The experiments cover a comprehensive range of tasks, datasets, and baselines. The model shows promising results on most datasets.

- The authors provide theoretical support for using latent diffusion models in regression tasks and discuss the advantages of converting deterministic tasks into generation tasks.

**Weaknesses:**

- The model functionality described in the introduction is somewhat misleading. In lines 28-29, the authors claim to construct a graph model capable of solving all task types. However, the implementation does not use a single foundation model to accomplish all tasks. Since the proposed framework still cannot solve the feature misalignment problem, specific tasks require training specific encoders and decoders. This makes the introduction seem like an overclaim, as LGD is just a framework that can handle multiple tasks but includes many components that need task-specific training. From a training cost perspective, this is similar with using separate models for different tasks (and potentially more expensive due to the need for GT and encoder-decoder). It is noteworthy that the authors mention this drawback in the limitations section.

- The experiments lack details on training and potential unfairness.

(1) In Appendix C, the authors describe that the encoder $\mathcal{E}$ was pretrained with both unsupervised and supervised methods. However, the main text does not specify which datasets were used for the encoder's training and whether they are the same as those used for training the diffusion model or if additional datasets were used. For instance, in the Conditional generation task, if the encoder was supervised-trained on the QM9 dataset, the results in Table 2 would lack fairness. This is because other baselines do not have access to supervised training information when providing conditions, giving the encoder a clear advantage. The authors should clarify the training data details for the encoder.

(2) In line 762, the authors state that the latent diffusion model is shared across different tasks, but the paper does not provide details on what data this shared diffusion model was trained on. Was it trained on the entire dataset across all tasks? This also lacks detailed description.

- Some of the authors' statements are overclaimed or inaccurate:

(1) In the analysis of the unconditional generation task (QM9), the authors claim that generation based on 3D information is easier, highlighting their method's effectiveness. This premise is incorrect. First, RNN-based models [1], which generate based on SMILES, can achieve a validity of 99% and high uniqueness without 3D information. 3D generation actually targets a different need—conformation generation—related to molecular dynamics and has broader research and practical value, hence the use of 3D information. It is not that generation based on 3D information is easier. Moreover, the three metrics the authors compare are not the best indicators of molecular generation quality. I suggest the authors use more effective metrics such as QED and TPSA. Previous works, especially those focusing on molecular generation tasks, do not limit evaluations to these two metrics due to their limited persuasiveness.

(2) In line 767 of the appendix, the authors claim their work is the first to achieve in-context learning on graph pretrained models. In fact, there are several prior works on graph prompts, such as PRODIGY [2].

[1] Molecular de-novo design through deep reinforcement learning. Journal of Cheminformatics.
[2] PRODIGY: Enabling In-context Learning Over Graphs. NeurIPS 2023.

- The discussion in lines 111-112 about the capabilities of generative models lacks citations.

**Questions:**

- Did the authors use a shared pretrained diffusion model for different tasks? If so, what data was it trained on?

- What do the authors think about the feature misalignment problem in graph pretraining?

**Limitations:**

Please refer to the Weakness and Questions above.

---

> ### Author Rebuttal · Authors · 2024-08-07
>
> We sincerely thank the reviewers for the constructive comments and suggestions. We address your concerns as follows.
>
> 1. Clarification on the introduction section (weakness 1). The introduction section mainly states the motivation to build a graph foundation model, but we do not claim that we have already done so (even in lines 28-29). We will surely rewrite the paper to avoid misunderstanding. Actually, LGD is indeed a unified framework to handle multiple tasks. Although we do not train a single model currently (as mentioned in our paper), we can indeed build a single graph foundation model under the LGD framework. For example, we can use OFA [1] as the autoencoder, which is a single model that embeds graphs from various domains into a unified latent space through a powerful LLM-GNN mixture architecture. We can then train a diffusion model in this latent space to tackle tasks of all levels and all types, utilizing the unified formulation in our paper. We believe the unified formulation of our paper is a significant step and a necessary stage towards graph foundation models.
>
> 2. Clarification on experiments (weakness 2). Appendix C actually describes the potential to build a complete pretrain, finetune and in-context learning framework using LGD. The contents are mainly discussions on future work about LGD, where we have not fully implemented them. We will move Appendix C to the discussion and future work section in the revised version of our paper and make it clearer.
>
> * For your question 2.(1) regarding the conditional generation task on QM9, the encoder is pretrained through the reconstruction loss of nodes and edges as described in the main context, without targets as supervision signals. The diffusion model still generates the molecules, and we follow the prior practice to use another pretrained model to evaluate the properties of all generated molecules, so our method does not introduce any unfairness.
>
> In addition, we would like to emphasize that even the encoders are trained in a supervised manner, it still does not introduce unfairness. Denote the conditions (properties) as $y$ and the molecules as $x$, there is a ground truth distribution $p(x|y)$. In training we sample $x\sim p(x|y)$ and denote it as a probabilistic mapping $f(y)$. The traditional conditional generation methods model $\hat x:=\theta(y)$ with a supervision signal $f(y)$. In comparison, our latent diffusion model learns the distribution of latent representation $\mathcal E(\hat x, \hat y):=\theta(y)$ with a supervision signal $\mathcal E(x, y)=\mathcal E(f(y), y)$. We can observe that both schemes include a supervision signal containing $y$ for generative models, except that we use an encoder to better represent the desired property. This is actually an advantage: utilizing representation learning, latent diffusion can better learns conditional distribution without introducing unfairness.
>
> * For your questions 2.(2), we train a separate encoder/decoder and diffusion model for every dataset for now. However, encoders and diffusion models trained across multiple datasets may have better performance and generalization, which we leave as future work.
>
> 3. Clarification on some statements. We thank the reviewer for the advice. There are indeed some statements that can be refined - we will correct them in the revision.
>
> (1) It is true that 3D generation is not necessarily easier. Our recent experiments also find that 3D generation can sometimes be harder due to the difference in data distribution and internal network architectures. We will correct related discussions in the paper. We also acknowledge the practical value of 3D generation and leave it as future work. Regarding the evaluation metrics, we additionally introduce the persuasive FCD and NSPDK metrics in our new experiments for better evaluation, and LGD achieves highly competitive performance. We do not include QED and TPSA for the QM9 dataset, since drug-likeliness is not suitable for the small molecules in QM9.
>
> (2) We are aware of PRODIGY and did not mention it since it works in a slightly different manner compared with the in-context learning in NLP: PRODIGY works by measuring the "similarity" between test and context examples, which is different from the conditional generation as in NLP. However, it is true that PRODIGY is also one sort of ICL, and we will make sure to include this citation and discussion in the revised version.
>
> 4. We will add the discussion concerning generalization ability of generative models, including LLM (Llama-3 and GPT-4 etc.) and diffusion models [2-3].
>
> 5. Answers to questions. (1) We train diffusion models separately as explained in 2. (2) Feature misalignment is indeed one of the obstacles to build a graph foundation model. LGD provides the potential simultaneously perform generation and regression/classification as long as graphs with different features are mapped to a unified latent space. Actually, OFA [1] is a single model that embeds graphs from various domains into a unified latent space through a powerful LLM-GNN mixture architecture. Combining LGD and OFA, we can then train a generative model in the latent space to overcome the difficulty of feature misalignment and task unification problems, which is left for future work. Overall, we believe that LGD is a significant contribution to the community and could inspire future work on graph foundation models.
>
> [1] Liu, H., Feng, J., Kong, L., Liang, N., Tao, D., Chen, Y., and Zhang, M. One for all: Towards training
> one graph model for all classification tasks. ICLR, 2024.
>
> [2] Zahra Kadkhodaie, Florentin Guth, Eero P Simoncelli, and Stephane Mallat. Generalization in diffusion models arises from geometry-adaptive harmonic representations. ICLR, 2024.
>
> [3] Puheng Li, Zhong Li, Huishuai Zhang, and Jiang Bian. On the generalization properties of diffusion models. NeurIPS, 2023.

---

> > ### Comment · Reviewer_XBTL · 2024-08-13
> >
> > Thanks for the response. I've read through other reviewers' feedback and responses as well. Most of my concerns have been addressed. Therefore, I will increase my rating to `6`. The authors should implement the revision promised in their response.

---

> > > ### Author Response · Authors · 2024-08-13
> > > **Thank you for your reply**
> > >
> > > We sincerely thank the reviewer for the kind reply. We will definitely include these results in the revision.

---

### Official Review · Reviewer_suUP · 2024-07-13

**Soundness:** 2
**Presentation:** 3
**Contribution:** 2
**Rating:** 5
**Confidence:** 3

**Summary:**

This paper presents Latent Graph Diffusion (LGD), a graph generation framework adept at handling a variety of tasks including generation, regression, and classification at different objects—node, edge, and graph levels. LGD approaches regression and classification tasks as conditional generation challenges and employs a latent diffusion model within the latent space to concurrently generate features for nodes, edges, and graphs.

**Strengths:**

1. The paper introduces the Latent Graph Diffusion as a novel approach to address the challenges of using a unified framework to achieve classification, regression and generation tasks on graph.

2. This paper clearly shows its idea.

**Weaknesses:**

1. While GDSS and DiGress are designed to handle various types of graph data beyond molecular diffusion, it is unclear if LGD has conducted similar graph generation experiments. The article lacks relevant general graph generation experiments and evaluations using metrics like MMD (Maximum Mean Discrepancy) to assess the effectiveness of graph generation.

2. The authors have not included some successful practices of diffusion models. Recent studies [1,2] have demonstrated better performance than the proposed method on the QM9 dataset.

3. The QM9 dataset used for graph generation is relatively small, and it would be beneficial for the authors to validate their proposed model on larger benchmark datasets, such as Moses [3], which also provide a wider range of evaluation metrics.

4. The description of the model's architecture details is unclear and could benefit from further clarification.

5. The submission does not include an open-source implementation of the proposed model.

[1] DiGress: Discrete denoising diffusion for graph generation. ICLR2022.
[2] Graph generation with diffusion mixture., ICML 2024.
[3] Molecular sets (moses): a benchmarking platform for molecular generation models. 2020.

**Questions:**

1.	It would be valuable to compare the computational efficiency of the evaluated methods during training and sampling to provide a comprehensive analysis.

**Limitations:**

N/A.

---

> ### Author Rebuttal · Authors · 2024-08-07
>
> We thank the reviewer for the useful comments and suggestions. As presented in the global response, we have conducted extensive new experiments to improve the paper. Here we address your concerns as follows.
>
> 1. General graph generation (weakness 1). As explained in our paper, LGD is capable of generating arbitrary graphs through the powerful continuous latent space, hence is able to handle various types of graph data including generic and synthetic graphs. We conduct generic graph generation experiments and report the results in the global response. We use SBM, Planar and Community-20 datasets for evaluation. LGD performs better than most traditional graph generation methods on these datasets (although it is not always the best). However, other methods often have hard set rules or structural information in the diffusion process, while LGD performs a pure diffusion process in the Euclidean latent space without input structural information, requiring the encoder to be powerful enough to encode structural information. While MPNN and graph transformers are known to have limited expressivity, they may fail to encode enough structural information like degree, clusters, orbits and eigenvalues. Moreover, the autoencoder is trained through reconstruction loss, which has limited supervision on high-order structures. Therefore, if expressive higher-order GNNs or proper structural encodings are used, or more pretraining tasks related to complex structures are adopt, we will be able to obtain a more powerful encoder and thus better generation qualities, which we leave as future work. Please note that LGD has superior performance in real-world tasks where node/edge features are often more important than high-order structures, making it easier to obtain more powerful and suitable encoders for diffusion.
>
> 2. Baselines and metrics (weakness 2). We rerun our experiments on QM9 and compare our results with state-of-the-art methods including DiGress [1], HGGT [2] and GruM [3]. We also include the FCD and NSPDK metrics to better evaluate the quality of generation. The results are shown in Table 1 of the pdf in the global response. In our new experiments, we train a more powerful encoder/decoder by incorporating positional encodings and more complex reconstruction tasks to obtain better latent graph representations, and carefully choose the hyper-parameters of the diffusion model. It is remarkable that our new results now are highly competitive or superior than the baselines in nearly all metrics. Especially, it achieves a better FCD score (in favor of generating graphs similar to the training data) than the three strong recent baselines [1,2,3] while sacrificing the least novelty.
>
> 3. Larger datasets (weakness 3). We are conducting new experiments on the larger benchmark dataset MOSES. The results are not yet ready due to the limited time. We will update the results as soon as possible, using comments and the link in the global response. We will include the new results in our revised paper. It is also notable that most recent studies including HGGT [2] and GruM [3] also do not include these datasets at scale.
>
> 4. Model architecture details (weakness 4). Actually, we have included our model architecture details in the appendix of our submission. Please refer to Appendix D.1 for our architecture design and implementation details, and Appendix D.3.1 for the detailed architectures utilized in each experiment.
>
> 5. Open-source implementation (weakness 5). We provide the code through the anonymous link in the global response. We will definitely include the open-source code in the final version of our paper.
>
> 6. Computation efficiency (question 1). We have actually included comprehensive analysis on computation efficiency in Appendix E.2 of our submission. In summary, the complexity of LGD depends on the specific architecture of the autoencoder and the denoising network, which is generally $O(n^2)$ for generation tasks and $O(n)$ for regression/classification tasks. The training stage of the diffusion model is fast, and the inference stage can also be accelerated through various methods (such as DDIM, which we already implemented). As a reference, training LGD requires about $0.2$s/epoch on Cora and about $16$s/epoch on ZINC.
>
> [1] C. Vignac, I. Krawczuk, A. Siraudin, B. Wang, V. Cevher, and P. Frossard. DiGress: Discrete denoising diffusion for graph generation. In International Conference on Learning Representations, 2022.
>
> [2] Y. Jang, D. Kim, and S. Ahn. Graph generation with K2-trees. In The Twelfth International Conference on Learning Representations, 2023.
>
> [3] Jaehyeong Jo, Dongki Kim, and Sung Ju Hwang. Graph generation with diffusion mixture., ICML 2024.

---

> > ### Comment · Reviewer_suUP · 2024-08-12
> > **Thank you for your response.**
> >
> > After reading the response, I decide to raise my score.
> > The additional result should be included in the revised version.

---

> > > ### Author Response · Authors · 2024-08-12
> > > **Thank you for your considered review and kind reply**
> > >
> > > We sincerely appreciate the reviewer for the kind reply. We will surely include these new results in the revised version.

---

### Author Rebuttal · Authors · 2024-08-07

We sincerely thank the reviewers for their insightful comments and suggestions. To address the concerns, we conduct new experiments as follows.

* For QM9 generation task, we (i) add more persuasive evaluation metrics including FCD and NSPDK to better evaluate the generation quality; (ii) take recent state-of-the-art baselines into account; and (iii) rerun our experiments with better configurations. In detail, we pretrain a more powerful encoder which takes nove-level and edge-level positional encodings (PE) as inputs, which is trained through more complex reconstruction tasks (decoding bond type with two connected atoms and decoding atom type with connected bonds). We then train a larger diffusion model with carefully searched hyper-parameters. Our new model achieves highly competitive and even state-of-the-art performance in all metrics except novelty. The convincing results validate the strong generation capability of LGD. Detailed results are shown in the pdf.

* We conduct generic graph generation experiments and report the results in the attached PDF. We use SBM, Planar and Community-20 datasets for evaluation. LGD performs better than most traditional graph generation methods on these datasets (although it is not always the best). However, other methods often have hard-set rules or additional structural information in the generation process, while LGD performs a pure diffusion process in the Euclidean latent space without explicitly input structures or hard rules, which requires the encoder to be very powerful to distinguish subtle structural differences. Since MPNN and graph transformers are known to have limited expressivity, they may fail to encode some structural information like clusters, orbits and eigenvalues, resulting in lower generation quality in pure-structure tasks. Please note that LGD has superior performance in real-world tasks where node/edge features are often as important as structures, making it easier to obtain more powerful and suitable encoders for diffusion. Overall, LGD is designed to handle various tasks on real-world graphs as shown by the other parts of our experiments.

* We are conducting experiments on larger datasets including MOSES. Due to the limited time, the final results are not yet ready. We will update the results as soon as the experiments are finished, through both official comments and the anonymous link below: https://anonymous.4open.science/r/NeurIPS2024Rebuttal-0290/.

* We also provide the code through the same anonymous [link](https://anonymous.4open.science/r/NeurIPS2024Rebuttal-0290/) above. We will release our source code in the final version.

---

> ### Author Response · Authors · 2024-08-12
> **Updated experimental results and we are looking forward to your reply**
>
> Dear reviewers,
> We now update the results on the large MOSES dataset for molecule generation at scale.
> $$
> \begin{array}{|cc|cccc|}
> \hline
> \text{Model} & \text{Class} & \text{Validity} & \text{Uniqueness} & \text{Novelty} & \text{FCD} \\\\
> \hline
> \text{VAE} & \text{SMILES} & 97.7 & 99.8 & 69.5 & 0.57\\\\
> \text{JT-VAE} & \text{Fragment} & 100 & 100 & 99.9 & 1.00\\\\
> \text{GraphINVENT} & \text{Autoreg.} & 96.4 & 99.8 & – & 1.22\\\\
> \text{ConGress} & \text{One-shot} & 83.4 & 99.9 & 96.4 & 1.48\\\\
> \text{DiGress} & \text{One-shot} & 85.7 & 100 & 95.0 & 1.19\\\\
> \hline
> \text{LGD (ours)} & \text{One-shot} & 95.9 & 100 & 95.9 & 1.42\\\\
> \hline
> \end{array}
> $$
> The results show that LGD has superior performance compared with DiGress in terms of validity and novelty metrics. Apart from DiGress and ConGress, we are the only model to scale to this large dataset, bridging the gap between diffusion-based one-shot graph generation model and other traditional methods like SMILES-based, fragment-based, and autoregressive models. Combining with other new results in our global response and the extensive experiments in the original submission, it is convincing that LGD reveals strong capability in both generation and prediction tasks.
>
> In addition, we sincerely appreciate reviewer suUP and a5WK's positive reply. We are looking forward to reviewer XBTL and PVEb's reply. We are happy to answer any further questions regarding the paper. Thank you.

---

### Decision · Program_Chairs · 2024-09-25

**Decision:**

Accept (poster)

**Comment:**

This paper presents a latent graph diffusion model which not only serves as a generative model, but it can also deal with classification and regression tasks at the node-, edge-, and graph-level.

While reviewers initially had some concerns and questions, most of them were addressed during the rebuttal phase, and all the reviewers are positive about this paper. The reviewers agree that the paper is generally well-written and that the idea of using a unified framework to solve classification, regression and generation tasks on graphs is novel. Indeed, while many of the ideas in this paper are not new (latent diffusion models have already been explored in the context of graphs [1]), unifying generation and prediction is an interesting direction. I would thus recommend the authors put more focus on this aspect. For example, the proposed method could be further evaluated in classification and regression tasks.

Overall, the consensus is that this is a useful contribution and should be accepted. I strongly advise the authors to include into the final version of the manuscript additional results and other revisions suggested by the reviewers and promised in the rebuttal.

[1] Xu, M., Powers, A. S., Dror, R. O., Ermon, S., & Leskovec, J. Geometric Latent Diffusion Models for 3D Molecule Generation. In ICML'23.